# Learning Equivariant Energy Based Models with Equivariant Stein Variational Gradient Descent

**Priyank Jaini**[*]
Bosch-Delta Lab
University of Amsterdam

**Lars Holdijk**[*]
Bosch-Delta Lab
University of Amsterdam

**Max Welling**
Bosch-Delta Lab
University of Amsterdam

## Abstract

We focus on the problem of efficient sampling and learning of probability densities by incorporating symmetries in probabilistic models. We first introduce *Equivariant Stein Variational Gradient Descent* algorithm – an equivariant sampling method based on Stein's identity for sampling from densities with symmetries. Equivariant SVGD explicitly incorporates symmetry information in a density through equivariant kernels which makes the resultant sampler efficient both in terms of sample complexity and the quality of generated samples. Subsequently, we define *equivariant* energy based models to model invariant densities that are learned using contrastive divergence. By utilizing our equivariant SVGD for training equivariant EBMs, we propose new ways of improving and scaling up training of energy based models. We apply these equivariant energy models for modelling joint densities in regression and classification tasks for image datasets, many-body particle systems and molecular structure generation.

## 1 Introduction

Many real-world observations comprise symmetries and admit probabilistic models that are invariant to such symmetry transformations. Naturally, overlooking these inductive biases while encoding such domains will lead to models with inferior performance capabilities. In this paper, we focus on the problem of efficient sampling and learning of equivariant probability densities by incorporating symmetries in probabilistic models.

We accomplish this by first proposing *equivariant Stein variational descent algorithm* in §3 for sampling from invariant densities. Stein Variational Gradient Descent (SVGD) is a kernel-based inference method that constructs a set of particles iteratively along an optimal gradient path in an RKHS to approximate and sample from a target distribution. We extend SVGD for invariant densities by considering equivariant kernel functions that evolve the set of particles such that the density at each time-step is invariant to the same symmetry transformations as encoded in the kernel. We demonstrate that equivariant SVGD is more sample efficient, produces a more diverse set of samples, and is more robust compared to regular SVGD when sampling from invariant densities.

Subsequently, in §4, we build *equivariant* Energy Based Models EBMs for learning invariant densities given access to i.i.d. data by leveraging the tremendous recent advances in geometric deep learning where the energy function is equivariant neural network. We train these equivariant EBMs through contrastive divergence by generating samples using equivariant SVGD. We show that incorporating the symmetries present in the data into the energy model as well as the sampler provides an efficient learning paradigm to train equivariant EBMs that generalize well beyond training data.

We empirically demonstrate the performance of equivariant EBMs using equivariant SVGD in §5. We consider real-world applications comprising of problems from many-body particle systems, molecular structure generation and, classification and generation for image datasets.

---

*Equal contribution.
35th Conference on Neural Information Processing Systems (NeurIPS 2021).

## 2 Preliminaries and Setup

In this section we set-up our main problem, introduce key definitions and notations and formulate an approach to incorporate symmetries in particle variational inference optimization methods through Stein variational gradient descent. Along the way, we also discuss directly related work and relegate a detailed discussion on previous work to Appendix B.

Let $\mathcal{G}$ be a group acting on $\mathbb{R}^d$ through a representation $\mathrm{R} : \mathcal{G} \to \mathrm{GL}(d)$ where $\mathrm{GL}(d)$ is the general linear group on $\mathbb{R}^d$, such that $\forall g \in \mathcal{G}$, $g \to \mathrm{R}_g$. Given a target random variable $\mathsf{X} \subseteq \mathbb{R}^d$ with density $\pi$, we say that $\pi$ is $\mathcal{G}$-invariant if $\forall g \in \mathcal{G}$ and $\boldsymbol{x} \in \mathbb{R}^d$, $\pi(\mathrm{R}_g \boldsymbol{x}) = \pi(\boldsymbol{x})$. Additionally, a function $f(\cdot)$ is $\mathcal{G}$-equivariant if $\forall g \in \mathcal{G}$ and $\boldsymbol{x} \in \mathbb{R}^d$, $f(\mathrm{R}_g \boldsymbol{x}) = \mathrm{R}_g f(\boldsymbol{x})$. We denote with $\mathcal{O}(\boldsymbol{x})$ the orbit of an element $\boldsymbol{x} \in \mathsf{X}$ defined as $\mathcal{O}(\boldsymbol{x}) := \{\boldsymbol{x}' : \boldsymbol{x}' = \mathrm{R}_g \boldsymbol{x}, \forall g \in \mathcal{G}\}$. We call $\pi_{|\mathcal{G}}$ the factorized density of a $\mathcal{G}$-invariant density $\pi$ where $\pi_{|\mathcal{G}}$ has support on the set $\mathsf{X}_{|\mathcal{G}}$ where the elements of $\mathsf{X}_{\mathcal{G}}$ are indexing the orbits i.e. if $\boldsymbol{x}, \tilde{\boldsymbol{x}} \in \mathsf{X}_{\mathcal{G}}$ then $\boldsymbol{x} \neq \mathrm{R}_g \tilde{\boldsymbol{x}}, \forall g \in \mathcal{G}$. In this paper, we are interested to incorporate *inductive biases* given by symmetry groups to develop efficient sampling and learning paradigms for generative modelling. Precisely, we consider the following problems:

**(i) Equivariant Learning:** Given access to an i.i.d. samples $\lfloor \boldsymbol{x}_1, \ldots, \boldsymbol{x}_n \rfloor \sim \pi$ from a $\mathcal{G}$-invariant density $\pi$, we want to approximate $\pi$. Rezende et al. (2019) and Köhler et al. (2020) addressed this by learning an equivariant normalizing flow (Tabak and Vanden-Eijnden, 2010; Tabak and Turner, 2013; Rezende and Mohamed, 2015) that transforms a simple latent $\mathcal{G}$-invariant density $q_0$ to the target density $\pi$ through a series of $\mathcal{G}$-equivariant diffeomorphic transformations $\mathbf{T} = (\mathbf{T}_1, \mathbf{T}_2, \cdots, \mathbf{T}_k)$ i.e. $\pi := \mathbf{T}_{\#} q_0$. They achieved this by proving (cf. (Köhler et al., 2020, Theorem 1), (Rezende et al., 2019, Lemma1)) that if $q_0$ is a $\mathcal{G}$-invariant density in $\mathbb{R}^d$, $\mathcal{F}$ is a proper sub-group of $\mathcal{G}$ i.e. $\mathcal{F} < \mathcal{G}$, and $\mathbf{T}$ is an $\mathcal{F}$-equivariant diffeomorphic transformation, then $\pi := \mathbf{T}_{\#} q_0$ is $\mathcal{F}$-invariant. However, a major drawback of this formalism is that it requires $\mathbf{T}$ to not only be a $\mathcal{F}$-equivariant diffeomorphism, but computation of the inverse and Jacobian must be cheap as well. This is problematic in practice.

Köhler et al. (2020) overcame this issue by using continuous normalizing flows (Grathwohl et al., 2018) that define a dynamical system through a time-dependent Lipschitz velocity field $\Psi : \mathbb{R}^d \times \mathbb{R}_+ \to \mathbb{R}^d$ with the following system of ordinary differential equations(ODEs):

$$\frac{\mathrm{d}\boldsymbol{x}(t)}{\mathrm{d}t} = \Psi(\boldsymbol{x}(t), t), \qquad \boldsymbol{x}(0) = \boldsymbol{z} \tag{1}$$

This allows to define a bijective function $\mathbf{T}_{\Psi,t}(\boldsymbol{z}) := \boldsymbol{x}(0) + \int_0^t \Psi(\boldsymbol{x}(t), t) \, \mathrm{d}t$ which leads to a push-forward density $q_t$ at each time-step $t$ satisfying $\frac{\mathrm{d} \log q_t}{\mathrm{d}t} = -\mathsf{div}\big(\Psi(\boldsymbol{x}(t), t)\big)$, which implies to the following important result:

**Lemma 1** ((Köhler et al., 2020, Theorem 2)). *Let $\Psi$ be an $\mathcal{F}$-equivariant vector-field on $\mathbb{R}^d$. Then, the transformation $\mathbf{T}_{\Psi,t}(\boldsymbol{z}) := \boldsymbol{x}(0) + \int_0^t \Psi(\boldsymbol{x}(t), t) \, \mathrm{d}t$ is $\mathcal{F}$-equivariant $\forall t \in \mathbb{R}_+$. Furthermore, the push-forward $q_t := \mathbf{T}_{\Psi,t,\#} q_0$ is $\mathcal{F}$-invariant $\forall t$, if $q_0$ is $\mathcal{G}$-invariant and $\mathcal{F} < \mathcal{G}$.*

Lemma 1 conveniently provides a framework to transform any $\mathcal{G}$-invariant density to an $\mathcal{F}$-invariant density along a path in which each intermediate density is also $\mathcal{F}$-invariant. However, equivariant normalizing flows cannot be used directly to generate samples when given access to an invariant density $\pi$ since they require i.i.d. samples from $\pi$ to train the flow[1].

**(ii) Equivariant Sampling:** In this paper, we are also interested in solving the inference problem i.e. we are interested in evaluating $\mathbb{E}_\pi[f]$, the expectation of $f$ when given access to a $\mathcal{G}$-invariant density $\pi$ which typically involves generating samples $\lfloor \boldsymbol{x}_1, \boldsymbol{x}_2, \cdots, \boldsymbol{x}_n \rfloor \sim \pi$. Intuitively, sampling from a $\mathcal{G}$-invariant density can be reduced to sampling from its corresponding factorized distribution $\pi_{|\mathcal{G}}$. This is because any set of samples $\{\tilde{\boldsymbol{x}}_i\}_{i=1}^n \sim \pi_{|\mathcal{G}}$ can be used to get samples representing $\pi$ by applying group actions from $\mathcal{G}$ to $\{\tilde{\boldsymbol{x}}_i\}_{i=1}^n$. Indeed, sampling methods like Markov Chain Monte Carlo (MCMC) (Brooks et al., 2011) or Hybrid Monte Carlo (HMC) Neal et al. (2011) and their variants, in principle, can use this paradigm to sample from an invariant density $\pi$. However, MCMC methods for approximate posterior sampling are often slow and it still remains challenging to scale them up to big data settings. An alternate to MCMC methods for approximate posterior sampling is Stein Variational Gradient Descent (SVGD) (Liu and Wang, 2016) which is a particle optimization

---

[1]In Appendix A, we discuss a way to use equivariant normalizing flow for direct sampling given access to $\pi$.

variational inference method that combines the paradigms of sampling and variational inference for Bayesian inference problems.

In SVGD, a set of $n$ particles $\{\boldsymbol{x}_i\}_{i=1}^n \in \mathsf{X} \subseteq \mathbb{R}^d$ are evolved following a dynamical system to approximate the target (posterior) density $\pi(\boldsymbol{x}) \propto \exp(-\mathrm{E}(\boldsymbol{x}))$ where $\mathrm{E}(\cdot)$ is the energy function. This is achieved in a series of $T$ discrete steps that transform the set of particles $\{\boldsymbol{x}_i^0\}_{i=1}^n \sim q_0(\boldsymbol{x})$ sampled from a base distribution $q_0$ (e.g. Gaussian) at $t = 0$ using the map $\boldsymbol{x}^t = \mathbf{T}(\boldsymbol{x}) := \boldsymbol{x}^{t-1} + \varepsilon \cdot \Psi(\boldsymbol{x}^{t-1})$ where $\varepsilon$ is the step size and $\Psi(\cdot)$ is a vector field. $\Psi(\cdot)$ is chosen such that it maximally decreases the KL divergence between the push-forward density $q_t(\boldsymbol{x}) = \mathbf{T}_{\#}q_{t-1}(\boldsymbol{x})$ and the target $\pi(\boldsymbol{x})$.

If $\Psi$ is restricted to the unit ball of an RKHS $\mathcal{H}_k^d$ with positive definite kernel $k : \mathbb{R}^d \times \mathbb{R}^d \to \mathbb{R}$, the direction of steepest descent that maximizes the negative gradient of the KL divergence is given by:

$$\Psi_{q,\pi}^*(\boldsymbol{x}) := \underset{\Psi \in \mathcal{H}_k^d}{\arg\max} -\nabla_\varepsilon \mathsf{KL}\big(q||\pi\big)|_{\varepsilon \to 0} = \mathbb{E}_{\boldsymbol{x} \sim q}[\mathsf{trace}(\mathcal{A}_\pi \Psi(\boldsymbol{x}))], \qquad (2)$$

where $\mathcal{A}_\pi \Psi(\boldsymbol{x}) = \nabla_{\boldsymbol{x}} \log \pi(\boldsymbol{x}) \Psi(\boldsymbol{x})^\top + \nabla_{\boldsymbol{x}} \Psi(\boldsymbol{x})$ is the Stein operator. Thus, an iterative paradigm can be easily implemented wherein a set of particles $\{\boldsymbol{x}_1^0, \boldsymbol{x}_2^0, \cdots, \boldsymbol{x}_n^0\} \sim q_0$ are transformed to approximate the target density $\pi(\cdot)$ using the optimal update $\Psi_{q,\pi}^*(\boldsymbol{x}) \propto \mathbb{E}_{\boldsymbol{x}' \sim q}[\mathcal{A}_\pi k(\boldsymbol{x}', \boldsymbol{x})]$. Since $\mathcal{A}_\pi \Psi(\boldsymbol{x}) = \nabla_{\boldsymbol{x}}[\pi(\boldsymbol{x})\Psi(\boldsymbol{x})]/\pi(\boldsymbol{x})$ we have that $\mathbb{E}_{\boldsymbol{x} \sim \pi}[\mathcal{A}_\pi \Psi(\boldsymbol{x})] = 0$ for any $\Psi$ implying convergence when $q = \pi$. Replacing the expectation in the update with a Monte Carlo sum over the current set of particles that represent $q$ we get:

$$\boldsymbol{x}_i^{t+1} \leftarrow \boldsymbol{x}_i^t + \varepsilon \tilde{\Psi}^*(\boldsymbol{x}_i^t), \text{ where, } \tilde{\Psi}^*(\boldsymbol{x}_i^t) := \frac{1}{n} \sum_{j=1}^n \big( \underbrace{\nabla_{\boldsymbol{x}_j^t} k(\boldsymbol{x}_j^t, \boldsymbol{x}_i)}_{\text{repulsive force}} - \underbrace{k(\boldsymbol{x}_j^t, \boldsymbol{x}_i) \cdot \nabla_{\boldsymbol{x}_j^t} \mathrm{E}(\boldsymbol{x}_j^t)}_{\text{attractive force}} \big) \quad (3)$$

Stein variational gradient descent intuitively encourages diversity among particles by exploring different modes in the target distribution $\pi$ through a combination of the second term in Equation (3) which attracts the particles to high density regions using the score function and the repulsive force (first term) which ensures the particles do not collapse together. In the continuous time limit, as $\varepsilon \to 0$, Equation (3) results in a system of ordinary differential equations describing the evolution of particles $\{\boldsymbol{x}_1^0, \boldsymbol{x}_2^0, \cdots, \boldsymbol{x}_n^0\}$ according to $\frac{\mathrm{d}\boldsymbol{x}}{\mathrm{d}t} = \tilde{\Psi}^*(\boldsymbol{x})$.

Furthermore, as shown in Wang et al. (2019), geometric information using pre-conditioning matrices can be incorporated in Equation (3) by using matrix valued kernels (cf. Definition 2.3 (Reisert and Burkhardt, 2007)) leading to the following generalized form of SVGD (Wang et al., 2019):

$$\boldsymbol{x}_i^{t+1} \leftarrow \boldsymbol{x}_i^t + \frac{\varepsilon}{n} \sum_{j=1}^n \big( \nabla_{\boldsymbol{x}_j^t} \boldsymbol{K}(\boldsymbol{x}_j^t, \boldsymbol{x}_i) - \boldsymbol{K}(\boldsymbol{x}_j^t, \boldsymbol{x}_i) \cdot \nabla_{\boldsymbol{x}_j^t} \mathrm{E}(\boldsymbol{x}_j^t) \big), \qquad (4)$$

where $\boldsymbol{K}(\boldsymbol{x}, \boldsymbol{x}')$ is a matrix valued kernel. Matrix-valued SVGD allows to flexibly incorporate preconditioning matrices yielding acceleration in the exploration of the given probability landscape.

SVGD has gained a lot of attention over the past few years as a flexible and scalable alternative to MCMC methods for approximate Bayesian posterior sampling. Further, it is more particle efficient since it generates diverse particles due to the deterministic repulsive force induced by kernels instead of Monte Carlo randomness. A natural question to ask is: *Can we incorporate symmetry information into SVGD for more efficient sampling from invariant densities?* We answer this in the affirmative in the next section by proposing *equivariant Stein variational gradient descent* algorithm for sampling from invariant densities.

## 3 Equivariant Stein Variational Gradient Descent

We begin this section by presenting the main result of this section by introducing *equivariant* Stein variational gradient descent (E-SVGD) by utilizing Lemma 1 and Equations (3) and (4).

**Proposition 1.** *Let $\pi$ be a $\mathcal{G}$-invariant density and $\lceil \boldsymbol{x}_1^0, \boldsymbol{x}_2^0, \cdots, \boldsymbol{x}_n^0 \rfloor \sim q_0$ be a set of particles at $t = 0$ with $q_0$ being $\mathcal{F}$-invariant where $\mathcal{F} > \mathcal{G}$. Then, the iterative update given by Equation (3) is $\mathcal{G}$-equivariant and the density $q_{t+1}$ defined by it at time $t + 1$ is $\mathcal{G}$-invariant if the positive definite kernel $k(\cdot, \cdot)$ is $\mathcal{G}$-invariant. The same holds for Equation (4) if $\boldsymbol{K}(\cdot, \cdot)$ is $\mathcal{G}$-equivariant.*

*Proof.* Since the initial distribution $q_0$ is $\mathcal{F}$-invariant, following Lemma 1 the update in Equation (3) is $\mathcal{G}$-equivariant if $\Psi$ is $\mathcal{G}$-equivariant. If $k(\cdot, \cdot)$ is $\mathcal{G}$-invariant then $\nabla_{\boldsymbol{x}} k(\cdot, \boldsymbol{x})$ is $\mathcal{G}$-equivariant. Furthermore, since $\pi = \exp\big(-\mathrm{E}(\boldsymbol{x})\big)$ is $\mathcal{G}$-invariant, $\nabla_{\boldsymbol{x}} \mathrm{E}(\boldsymbol{x})$ is also $\mathcal{G}$-equivariant. Thus, both the terms for $\Psi$ are $\mathcal{G}$-equivariant if $k(\cdot, \cdot)$ is $\mathcal{G}$-equivariant making the update in Equation (3) $\mathcal{G}$-equivariant. The result follows similarly for Equation (4) when $\boldsymbol{K}(\cdot, \cdot)$ is $\mathcal{G}$-equivariant. $\qquad\square$

Following Proposition 1, we call the updates in Equations (3) & (4) *equivariant* Stein variational gradient descent when the kernel $k(\cdot, \cdot)$ (and $\boldsymbol{K}(\cdot, \cdot)$ respectively) is invariant (equivariant) and the initial set of particles $\lbrace \boldsymbol{x}_1^0, \cdots, \boldsymbol{x}_n^0 \rbrace$ are sampled from an invariant density $q_0$. Thus, all that is required to sample from a $\mathcal{G}$-invariant density $\pi$ using equivariant SVGD is to construct a positive definite kernel that is $\mathcal{G}$-equivariant. Let us next give a few examples for constructing in- and equivariant positive definite kernels.

**Example 1** (Invariant scalar kernel)**.** *Let $\mathcal{G}$ be a finite group acting on $\mathbb{R}^d$ with representation $\mathrm{R}$ such that $\forall g \in \mathcal{G}, g \to \mathrm{R}_g$. Then,*

$$k_{\mathcal{G}}(\boldsymbol{x}, \boldsymbol{x}') = \sum_{\boldsymbol{x} \in \mathcal{O}(\boldsymbol{x})} \sum_{\boldsymbol{x}' \in \mathcal{O}(\boldsymbol{x}')} k(\boldsymbol{x}, \boldsymbol{x}')$$

*is $\mathcal{G}$-invariant where $k(\cdot, \cdot)$ is some positive-definite kernel. While this provides a general method to construct invariant kernels for finite groups, the double summation can be computationally expensive. In practice, usually simple kernels like RBF kernel (for rotation symmetries) or uniform kernel suffice as more practical alternatives.*

Example 1 is only restricted to finite groups and does not directly apply to continuous symmetry groups. We can construct kernels for continuous groups following Example 1 by either using a Monte Carlo approximation or using a transformation that performs computations in the factorized space $\mathsf{X}_{|\mathcal{G}}$ as we show in the next example.

**Example 2** (Continuous Symmetry Groups)**.** *Let $\pi(\boldsymbol{x})$ be $\mathsf{SO}(2)$-invariant (cf. Figure 3b for an example) where $\boldsymbol{x} \in \mathbb{R}^2$ i.e. $\mathcal{O}(\boldsymbol{x}) := \{\boldsymbol{x}' : \|\boldsymbol{x}\| = \|\boldsymbol{x}'\|\}$. We can either construct an invariant kernel for sampling from $\pi$ using a Monte Carlo approximation by sampling random rotations on a unit sphere i.e.*

$$k_{\mathcal{G}}(\boldsymbol{x}, \boldsymbol{x}') = \sum_{i,j=1}^n k(g_j \boldsymbol{x}, g_i \boldsymbol{x}'), \qquad g_i, g_j \in \mathcal{G}, \ \forall (i, j) \in [n] \times [n]$$

*Or alternately, we can consider the function $\Phi_{\mathcal{G}} : \mathbb{R}^2 \to \mathbb{R}$ such that $\Phi_{\mathcal{G}}(\boldsymbol{x}) = \|\boldsymbol{x}\|$. Then, $\Phi_{\mathcal{G}}(\boldsymbol{x})$ is $\mathsf{SO}(2)$ invariant since $\Phi_{\mathcal{G}}(g\boldsymbol{x}) = \Phi_{\mathcal{G}}(\boldsymbol{x}), \forall g \in \mathcal{G}$. Thus, we can now use the following kernel*

$$k_{\mathcal{G}}(\boldsymbol{x}, \boldsymbol{x}') = k\big(\Phi_{\mathcal{G}}(\boldsymbol{x}), \Phi_{\mathcal{G}}(\boldsymbol{x}')\big)$$

Examples (1) and (2) are both invariant scalar kernels. Let us next give an example of an equivariant matrix valued kernel for matrix valued SVGD (cf. Equation (4)).

**Example 3** (Equivariant Matrix-Valued Kernels, Reisert and Burkhardt (2007))**.** *Examples 1 and 2 define an invariant scalar kernel. Following Reisert and Burkhardt (2007), we can also construct a $\mathcal{G}$-equivariant matrix-valued kernel for the generalized update as in Equation (4) by defining:*

$$\boldsymbol{K}(\boldsymbol{x}, \boldsymbol{x}') = \int_{\mathcal{G}} k(\boldsymbol{x}, g\boldsymbol{x}') \mathrm{R}_g \, \mathrm{d}g$$

*where $\mathrm{R}_g$ is a group representation and $k(\cdot, \cdot)$ is a scalar symmetric, $\mathcal{G}$-invariant function. It is easy to check that $\boldsymbol{K}(\boldsymbol{x}, \boldsymbol{x}')$ is equivariant in the first argument and anti-equivariant in the second argument, leading to an equivariant $\boldsymbol{K}(\boldsymbol{x}, \boldsymbol{x}')$ (cf. Proposition 2.2 Reisert and Burkhardt (2007)).*

**Advantages of Equivariant Sampling:** As we discussed briefly in Section 2, SVGD works by evolving a set of particles using a dynamical system through a combination of attractive and repulsive forces among the particles that are governed by the inter-particle distance. Thus, a particle exerts these forces in a restricted neighbourhood around it. Equivariant SVGD, on the other hand, is able to model *long-range interactions* among particles due to the use of equivariant kernel. Intuitively, any point $\boldsymbol{x}$ is able to exert these forces on any other point $\boldsymbol{x}'$ in equivariant SVGD if $\boldsymbol{x}'$ is in the

neighbourhood of any point in the orbit $\mathcal{O}(\boldsymbol{x})$ of $\boldsymbol{x}$. This is because for any point $\boldsymbol{x}'$ the repulsive and attractive force terms are the same in Equations (3) and (4) for all points that are in the orbit $\mathcal{O}(\boldsymbol{x})$. This ability to capture long-range interactions by equivariant Stein variational gradient descent subsequently makes it more efficient in sample complexity and running time with better sample quality, and makes it more robust to different initial configurations of the particles compared to vanilla SVGD. We illustrate these next with the help of the following examples:

**(i) $C_4$-Gaussians** (cf. Figure 3a and 3c): This example consists of four Gaussians invariant to $C_4$ symmetry group. In this case, the group factorized distribution $\pi_{|C_4}$ is Gaussian with the original $C_4$-invariant density obtained by rotating $\pi_{|C_4}$ through the set $\{0°, 90°, 180°, 270°\}$. In Figure 3a, the first column shows the samples generated by equivariant SVGD, the second column is the projection of these samples on the group factorized space $\mathsf{X}_{|C_4}$ and, the third column shows the samples obtained by rotating the original samples through the $C_4$-symmetry group. Figure 3c shows a similar setup for vanilla SVGD.

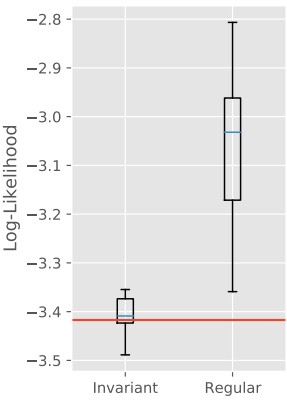

Figure 1: Sample efficiency

**(ii) Concentric Circles** (cf. Figure 3b and 3d): This example comprises of two concentric circles invariant to the $\mathsf{SO}(2)$ symmetry group. In this case, the group factorized space is a union of two disconnected lines with length equal to the thickness of the circles. In Figure 3b, the first column shows the samples generated by equivariant SVGD and, the second column is the projection of these samples on the group factorized space $\mathsf{X}_{|\mathsf{SO}(2)}$. Figure 3d shows a similar setup for vanilla SVGD.

We keep the experimental setup i.e. number of particles and number iterations exactly the same for both vanilla SVGD and equivariant SVGD. For both the examples, it may seem that the original samples from the vanilla SVGD capture the target distribution better than the equivariant counterpart (first column for Figs. 3a-3d). However, projecting the samples onto the factorized space (second column for the aforementioned figures) shows that equivariant SVGD more faithfully captures the target density compared to vanilla SVGD. Furthermore, due to its ability to model long-range interactions, we see for both examples that in the projected space of the invariant sampler, the samples are not close together whereas for vanilla

Figure 2: Robustness

SVGD most samples end up in a configuration where they reside in the same orbit. This phenomena is most evident for the concentric circles example where samples from vanilla SVGD reside on the high density region throughout the two circles resulting in all the samples being positioned on top of each other in the factorized space demonstrating its inability to capture the distribution. On the other hand, invariant SVGD prevents any sample from residing on the same orbit of another sample due to long-range repulsive force from the equivariant kernel allowing it to sample more faithfully from the invariant densities.

Secondly, we study the effect of increasing the number of particles used for vanilla SVGD for the two concentric circles example. In Figure 1, we plot the average log-likelihoods of the particles from vanilla SVGD and particles from invariant SVGD as a function of number of iterations and compare it to the ground-truth average log-likelihood. We run vanilla SVGD with up to 32 times more particles than invariant SVGD. As evident from the plot, invariant SVGD converges to the final configuration within the first 100 iterations with average log-likelihood closely matching the ground truth. Vanilla SVGD, on the other hand, is unable to converge to the ground truth with even 32 times more samples and 5000 iterations due to its inability to interact with particles at longer distances.

Finally, we study the effect of different configurations of the initial particles on the performance of vanilla and invariant SVGD in Figure 2 for the $C_4$-Gaussian example. As shown by Zhuo et al. (2018); Zhang et al. (2020) and D'Angelo and Fortuin (2021), the particles in vanilla SVGD have a tendency

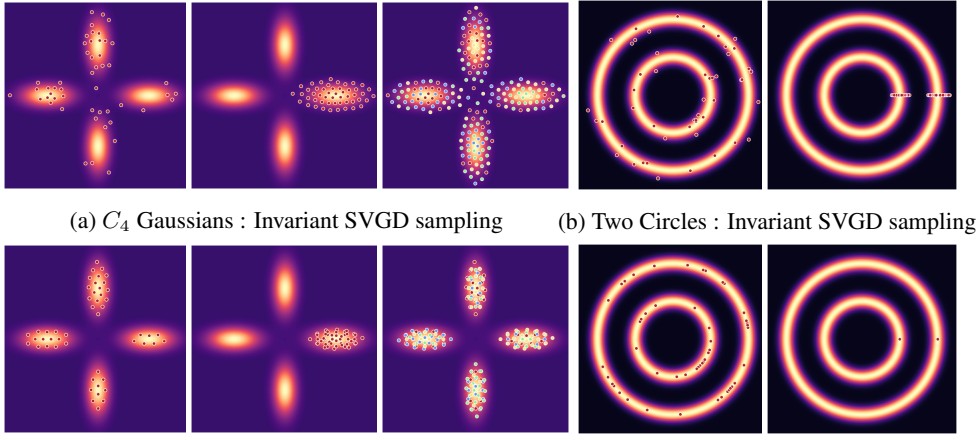

(a) $C_4$ Gaussians : Invariant SVGD sampling      (b) Two Circles : Invariant SVGD sampling

(c) $C_4$ Gaussians : Vanilla SVGD Sampling      (d) Two Circles : Vanilla SVGD Sampling

Figure 3: *Recommended to view in color*. 3a (Left to Right) Original Samples from E-SVGD, samples projected on to the group-factorized space and, samples obtained after applying group actions to the original samples. Yellow, Green and, Blue samples represent original samples rotated by $90°$, $180°$ and, $270°$ respectively. 3b (Left to Right) Original Samples from E-SVGD and, samples projected on to the group-factorized space. 3c-3d: Same as 3a-3b but for vanilla SVGD.

to collapse to a few local modes that are closest to the initial distribution of the particles. We test the robustness of invariant SVGD to particles with initial distributions localized to different regions in the space. We plot the average log-likelihoods of the converged samples for both invariant and vanilla SVGD for all random initializations in Figure 2 and compare this to the ground truth average log-likelihood. The plot illustrates that equivariant SVGD is more robust to the initial distribution of particles than vanilla SVGD. Nevertheless, if the group-factorized space is multi-modal, equivariant SVGD might exhibit a tendency to favour one of modes. However, this can be easily alleviated by either adding some noise to the SVGD update as proposed by Zhang et al. (2020) similar to SGLD (Welling and Teh, 2011) or using an annealing strategy (D'Angelo and Fortuin, 2021).

## 4 Equivariant Joint Energy Model

In Section 3, we developed equivariant Stein variational gradient descent algorithm for sampling from invariant densities. In this section, we leverage the recent tremendous advances in deep geometric learning (Cohen and Welling, 2016; Dieleman et al., 2016; Bronstein et al., 2021) to propose *equivariant energy based models* that are trained contrastively using our proposed equivariant Stein variational gradient descent algorithm to learn invariant (unnormalized) densities $\pi$ given access to i.i.d. samples $\{x_1, x_2, \cdots, x_n\} \sim \pi$.

Given a set of samples $\{x_1, x_2, \cdots, x_n\} \subseteq \mathbb{R}^d$, energy-based models (LeCun et al., 2006) learn an energy function $\mathrm{E}_{\boldsymbol{\theta}}(x) : \mathbb{R}^d \to \mathbb{R}$ that defines a probability distribution $\tilde{\pi}_{\boldsymbol{\theta}}(x) = \exp(-\mathrm{E}_{\boldsymbol{\theta}}(x))/Z_{\boldsymbol{\theta}}$, where $Z_{\boldsymbol{\theta}} = \int \exp(-\mathrm{E}_{\boldsymbol{\theta}}(x)) \, dx$ is the partition function. Unlike other popular tractable density models like normalizing flows, EBMs are less restrictive in the parameterization of the functional form of $\tilde{\pi}_{\boldsymbol{\theta}}(\cdot)$ since the energy function does not need to integrate to one, it can be parameterized using any nonlinear function. Conveniently, if $\pi$ is $\mathcal{G}$-invariant, we can use the existing equivariant deep network architectures to parameterize $\mathrm{E}_{\theta}(\cdot)$ to encode the symmetries into the energy network. Such an equivariant energy network defines an *equivariant* energy based model. EBMs are usually trained by maximizing the log-likelihood of the data under the given model:

$$\boldsymbol{\theta}^* := \arg\min_{\boldsymbol{\theta}} \mathcal{L}_{\mathsf{ML}}(\boldsymbol{\theta}) = \mathbb{E}_{\boldsymbol{x}\sim\pi}\big[ -\log \tilde{\pi}_{\boldsymbol{\theta}}(\boldsymbol{x})\big] \tag{5}$$

However, evaluating $Z_{\boldsymbol{\theta}}$ is intractable for most (useful) choices of $\mathrm{E}_{\boldsymbol{\theta}}(\cdot)$ which makes learning EBMs via maximum likelihood estimation problematic. Contrastive divergence (Hinton et al., 2006) provides a paradigm to learn EBMs using maximum likelihood estimation without needing to compute $Z_{\boldsymbol{\theta}}$ by

approximating the gradient of $\nabla_{\boldsymbol{\theta}} \mathcal{L}_{\mathsf{ML}}(\boldsymbol{\theta})$ in Equation (5) as follows:

$$\nabla_{\boldsymbol{\theta}} \mathcal{L}_{\mathsf{ML}}(\boldsymbol{\theta}) \approx \mathbb{E}_{\boldsymbol{x}^+ \sim \pi}\left[\nabla_{\boldsymbol{\theta}} \mathrm{E}_{\boldsymbol{\theta}}(\boldsymbol{x}^+)\right] - \mathbb{E}_{\boldsymbol{x}^- \sim \tilde{\pi}_{\boldsymbol{\theta}}}\left[\nabla_{\boldsymbol{\theta}} \mathrm{E}_{\boldsymbol{\theta}}(\boldsymbol{x}^-)\right] \tag{6}$$

Intuitively, the gradient in Equation (6) drives the model such that it assigns higher energy to the negative samples $\boldsymbol{x}^-$ sampled from the current model and decreases the energy of the positive samples $\boldsymbol{x}^+$ which are the data-points from the target distribution. Since, training an EBM using MLE requires sampling from the current model $\tilde{\pi}(\boldsymbol{\theta})$, successful training of EBMs relies heavily on sampling strategies that lead to faster mixing. Fortuitously, since $\mathrm{E}_{\boldsymbol{\theta}}(\cdot)$ in our present setting is $\mathcal{G}$-equivariant, we propose to use our equivariant sampler for more efficient training[2] of the equivariant energy based model.

Additionally, following Grathwohl et al. (2019), we can extend equivariant energy based models to equivariant joint energy models. Let $\{(\boldsymbol{x}_1, y_1), (\boldsymbol{x}_2, y_2), \cdots, (\boldsymbol{x}_n, y_n)\} \subseteq \mathbb{R}^d \times [K]$ be a set of samples with observations $\boldsymbol{x}_i$ and labels $y_i$. Given a parametric function $f_{\boldsymbol{\theta}} : \mathbb{R}^d \to \mathbb{R}^k$, a classifier uses the conditional distribution $\tilde{\pi}_{\boldsymbol{\theta}}(y|\boldsymbol{x}) \propto \exp(f_{\boldsymbol{\theta}}(\boldsymbol{x})[y])$ where $f_{\boldsymbol{\theta}}(\boldsymbol{x})[y]$ is the logit corresponding to

---

**Algorithm 1:** Equivariant EBM training

**Input:** $\{\boldsymbol{x}_1^+, \boldsymbol{x}_2^+, \cdots, \boldsymbol{x}_m^+\} \sim \pi(\boldsymbol{x})$
**while** *not converged* **do**
$\quad \triangleright$ *Generate samples from current eqNN model* $\mathrm{E}_{\boldsymbol{\theta}}$
$\quad \{\boldsymbol{x}_1^-, \boldsymbol{x}_2^-, \cdots, \boldsymbol{x}_m^-\} = \mathsf{EquivariantSVGD}(\mathrm{E}_{\boldsymbol{\theta}})$ ;
$\quad \triangleright$ *Optimize objective* $\mathcal{L}_{\mathsf{ML}}(\boldsymbol{\theta})$:
$\quad \Delta\boldsymbol{\theta} \leftarrow \sum_{i=1}^m \nabla_{\boldsymbol{\theta}} \mathrm{E}_{\boldsymbol{\theta}}(\boldsymbol{x}_i^+) - \nabla_{\boldsymbol{\theta}} \mathrm{E}_{\boldsymbol{\theta}}(\boldsymbol{x}_i^-)$ ;
$\quad \triangleright$ *Update* $\boldsymbol{\theta}$ *using* $\Delta\boldsymbol{\theta}$ *and Adam optimizer*
**end**

---

the $y^{\text{th}}$ class label. As shown by Grathwohl et al. (2019), these logits can be used to define the joint density $\tilde{\pi}_{\boldsymbol{\theta}}(\boldsymbol{x}, y)$ and marginal density $\tilde{\pi}_{\boldsymbol{\theta}}(\boldsymbol{x})$ as follows:

$$\tilde{\pi}_{\boldsymbol{\theta}}(\boldsymbol{x}, y) = \frac{\exp\big(f_{\boldsymbol{\theta}}(\boldsymbol{x})[y]\big)}{Z_{\boldsymbol{\theta}}}, \quad \text{and,} \quad \tilde{\pi}_{\boldsymbol{\theta}}(\boldsymbol{x}) = \frac{\sum_y \exp\big(f_{\boldsymbol{\theta}}(\boldsymbol{x})[y]\big)}{Z_{\boldsymbol{\theta}}} \tag{7}$$

Hence, the energy function at a point $\boldsymbol{x}$ is given by $\mathrm{E}_{\boldsymbol{\theta}} = -\log \sum_y \exp(f_{\boldsymbol{\theta}}(\boldsymbol{x})[y])$ with joint energy $\mathrm{E}_{\boldsymbol{\theta}}(\boldsymbol{x}, y) = -f_{\boldsymbol{\theta}}(\boldsymbol{x})[y]$. In our setting, the joint distribution $\pi(\boldsymbol{x}, y)$ is $\mathcal{G}$-invariant in the first argument i.e. $\pi(\mathrm{R}_g \boldsymbol{x}, y) = \pi(\boldsymbol{x}, y), \forall g \in \mathcal{G}$. An example of such a setting is any image data-set where the class label does not change if the image is rotated by an angle. Using Equation (7), it suffices for the function $f_{\boldsymbol{\theta}}$ to be $\mathcal{G}$-equivariant to model a $\mathcal{G}$-invariant density $\tilde{\pi}_{\boldsymbol{\theta}}(\boldsymbol{x}, y)$. Furthermore, a $\mathcal{G}$-equivariant $f_{\boldsymbol{\theta}}$ also makes the marginal density $\tilde{\pi}_{\boldsymbol{\theta}}(\boldsymbol{x})$ and conditional density $\tilde{\pi}_{\boldsymbol{\theta}}(y|\boldsymbol{x})$ $\mathcal{G}$-invariant in the input $\boldsymbol{x}$. We call such an energy model where $f_{\boldsymbol{\theta}}$ is equivariant to a symmetry transformation to be an *equivariant* joint energy model.

We can train this model by maximizing the log-likelihood of the joint distribution as follows:

$$\mathcal{L}(\boldsymbol{\theta}) := \mathcal{L}_{\mathsf{ML}}(\boldsymbol{\theta}) + \mathcal{L}_{\mathsf{SL}}(\boldsymbol{\theta}) = \log \tilde{\pi}_{\boldsymbol{\theta}}(\boldsymbol{x}) + \log \tilde{\pi}_{\boldsymbol{\theta}}(y|\boldsymbol{x}) \tag{8}$$

where $\mathcal{L}_{\mathsf{SL}}(\boldsymbol{\theta})$ is the supervised loss which is the cross-entropy loss in the case of classification. Thus, an equivariant joint energy model can now be trained by applying the gradient estimator in Equation (6) for $\log \tilde{\pi}_{\boldsymbol{\theta}}(\boldsymbol{x})$ and evaluating the gradient of $\log \tilde{\pi}_{\boldsymbol{\theta}}(y|\boldsymbol{x})$ through back-propagation. Conveniently, Equation (8) can also be used for semi-supervised learning with $\mathcal{L}_{\mathsf{SL}}((\boldsymbol{\theta}))$ substituted with the appropriate supervised loss e.g. MSE for regression.

Let us end this section with an empirical example for learning a mixture of $C_4$-Gaussians (Figure 4) as shown in row two of the leftmost column of Figure 4. The innermost $C_4$-Gaussian defines the class conditional probability $\pi(\boldsymbol{x}|y = 0)$ (row 3) and the outer $C_4$-Gaussian defines $\pi(\boldsymbol{x}|y = 1)$ (row 4). We learn a non-equivariant joint EBM using vanilla SVGD (cf. Figure 4 center column) and an equivariant joint EBM using equivariant SVGD (cf. Figure 4 right column) keeping the number of iterations and particles the same for training. In Figure 4, we plot the decision boundaries learned by the model in the top row. The star marked samples in the figure are the samples generated by the underlying model. We plot the joint distribution and the class conditional distributions in row two-four respectively. The figure abundantly demonstrates the superior performance of an equivariant joint energy model trained using equivariant SVGD over its non-equivariant counterpart. A more detailed figure with comparisons to an equivariant joint energy model trained using vanilla SVGD is presented in Appendix D.1.

---

[2]compared to using a regular sampler with no encoded symmetries.

# 5 Experiments

In this section, we present empirical analysis of equivariant EBMs and E-SVGD through experiments to (i) reconstruct potential function describing a many-body particle system (DW-4) trained using limited number of meta-stable states, (ii) model a generative distribution of molecules (QM9) and generate novel samples and (iii) hybrid (generative & discriminative) model invariant to rotations for FashionMNIST trained using dataset with no rotations. Due to space constraints, details about all the experiments as well as detailed figures are relegated to Appendix E.

**DW-4:** In this many-body particles system, a double-well potential describes the configuration of four particles that is invariant to rotations, translations and, permutation of the particles. This system comprises five distinct metastable states which are characterized as the mimina in the potential function. In our experiment, we show that given access to only a single example of each metastable state configuration, an equivariant EBM trained with E-SVGD can recover other states with similar energy as those of in the training set. In Figure 7, the first column shows the metastable states present in the training set. The second column are the states recovered by an EBM trained with vanilla SVGD which results in configurations that exactly copy the training set. The third column shows configurations generated by the equivariant model which are distinct from the training set but mimic the energies of the corresponding metastable states in the training set.

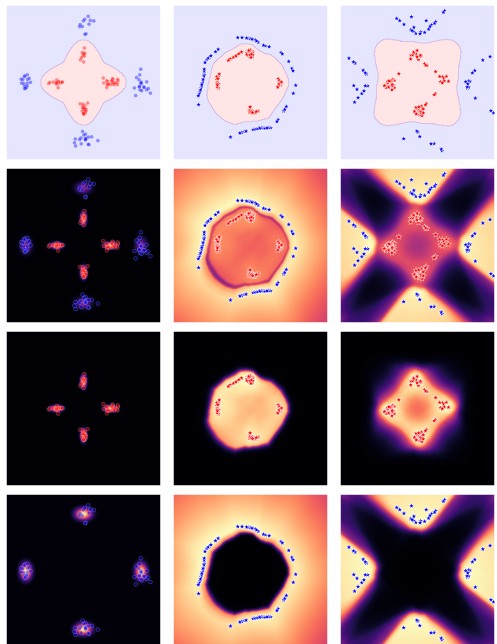

Figure 4: $C_4$-Gaussian mixture model. *Row 1:* Decision Boundary. *Row 2:* Samples and energy of joint distribution $\pi(\boldsymbol{x}, y)$. *Row 3:* Samples and energy of conditional distribution $\pi(\boldsymbol{x}|y = 0)$. *Row 4:* Samples and energy of conditional distribution $\pi(\boldsymbol{x}|y = 1)$. *Left:* Target distribution. *Middle:* Non-equivariant EBM trained with vanilla SVGD. *Right:* E-EBM trained with E-SVGD.

Our setup is different from that of Köhler et al. (2020); we discuss this in detail in Appendix E.1 and also produce similar results as Köhler et al. (2020) for our model.

**QM9:** QM9 is a molecular dataset containing over 145,000 molecules used for molecular property prediction. However, we use this for molecular structure generation of constitutional isomers of $C_5H_8O_1$. Similar to DW-4, the molecules here are invariant to rotations, translations and, permutations of the same atoms. We encode these symmetries using E-GNN (Satorras et al., 2021), an equivariant graph neural network, to represent the energy. We trained our model via E-SVGD using $C_5H_8O_1$ molecules present in the QM9 dataset and used the trained energy model to generate novel samples that are isomers of $C_5H_8O_1$. We show these novel generated molecules in Figure 5 wherein we used the relative distance between atoms as a proxy for determining the covalent bonds. Our generated

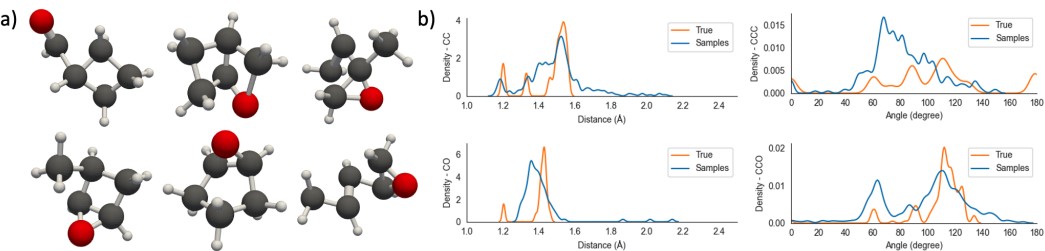

Figure 5: (a) Molecules sampled from a EBM parameterized by a E-GNN trained using E-SVGD. (b) Distribution of distance and angle between atom pairs and triplets.

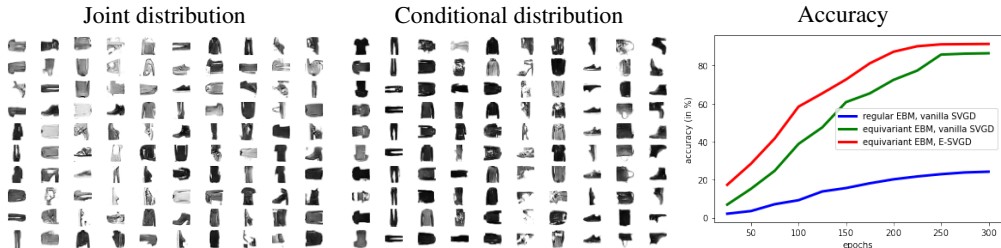

Joint distribution       Conditional distribution       Accuracy

Figure 6: *Left & Center:* Samples generated from joint and class-conditional distribution using equivariant EBM. *Right:* Plot of classification accuracy vs. training iterations for equivariant and regular EBMs trained using vanilla SVGD and E-SVGD.

molecules demonstrate the correct 3D arrangement of bonds while containing complex atom structures like aromatic rings. This is further supported by the plots comparing the radial distribution functions of the two most common heavy atom pairs to quantify our model fit to QM9 (Figure 5). While, the generated molecules have a larger distributional spread, the range of values and modes – for both angles and distances– resemble the true distribution. We provide more details in Appendix E.2.

**FashionMNIST:** (Details in Appendix E.3) In this experiment, we take the FashionMNIST dataset with training set consisting regular images whereas the test set is processed to contain images that are randomly rotated using the $C_4$-symmetry group. We train an equivariant energy model where the energy function is a $C_4$ steerable CNN (Weiler and Cesa, 2019) with both E-SVGD and vanilla SVGD. Furthermore, we also compare to an energy model with no rotation symmetries and depict the performance in terms of classification accuracy on the held out images of these three models as a function of the number of training iterations. The plot in Figure 6 shows, albeit unsurprisingly, that an equivariant energy model performs better than a regular model. Furthermore, the results also illustrate that an equivariant model trained with E-SVGD converges faster than when trained with vanilla SVGD highlighting the benefit of using E-SVGD for training equivariant EBMs. Furthermore, in Figure 6, we show samples generated by E-SVGD using the trained equivariant EBM from the joint and the class-conditional distribution.

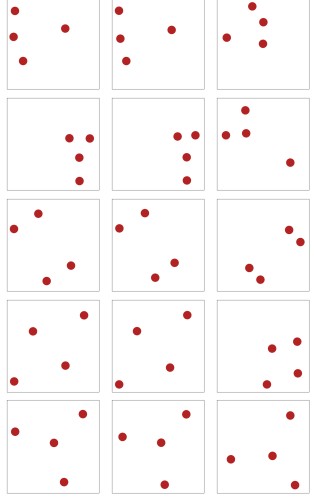

Figure 7: *Col. 1*: Samples from true potential energy. *Col. 2*: Samples from EBM trained with SVGD. *Col. 3*: Samples from equivariant EBM trained with E-SVGD.

## 6  Discussion and Conclusion

In this paper, we focused on incorporating *inductive bias* in the form of symmetry transformations using equivariant functions for sampling and learning invariant densities. We first proposed *equivariant* Stein variational gradient descent algorithm for sampling from invariant densities by using equivariant kernels which affords many benefits in terms of efficiency due to its ability to model long-range interactions between particles. However, a major limitation of Stein variational gradient descent algorithm in general is its sensitivity to the kernel hyper-parameters. An interesting future work might be to develop strategies to either adapt or learn these hyper-parameters while running the SVGD dynamics.

Subsequently, we proposed *equivariant* energy based models wherein the energy function is parameterized by an equivariant network. In our experiments, we leveraged the recent advances in geometric deep learning to model EBMs using steerable CNNs (Weiler and Cesa, 2019) for images, equivariant graph networks (Satorras et al., 2021) for representing molecules, and group equivariant networks (Cohen and Welling, 2016) for many-body particle systems. We used equivariant SVGD to train these equivariant energy based models for modelling invariant densities and demonstrated that incorporating symmetries in the energy model as well as the sampler leads to efficient training. However, as discussed in previous works (Grathwohl et al., 2019), training EBMs using contrastive divergence and short sampling chains is often unstable and challenging. These issues remain with equivariant samplers and have to be addressed to be able to train large-scale energy based models.

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
