# A Sampling using Equivariant Flows

Neural transport augmented sampling, first introduced by Parno and Marzouk (2018), is a general method for using normalizing flows to sample from a given density $\pi$. Informally, the method proceeds by learning a diffeomorphic map $\mathbf{T} : \mathsf{Z} \to \Theta$ such that $\tilde{p}(\mathbf{z}) = \pi(\boldsymbol{\theta}) \cdot |\mathbf{T}'(\mathbf{z})|$ where $\mathbf{z} = \mathbf{T}^{-1}(\boldsymbol{\theta})$ such that $p(\mathbf{z})$ has a simple geometry amenable to efficient MCMC sampling. Thus, samples can be generated from $\pi(\boldsymbol{\theta})$ by running MCMC chain in the Z-space and pushing these samples onto the $\Theta$-space using $\mathbf{T}$. The transformation $\mathbf{T}$ can be learned by minimizing the KL-divergence between a fixed distribution with simple geometry in the z-space e.g. a standard Gaussian and $\tilde{p}(\mathbf{z})$ above. The learning phase attempts to ensure that the distribution $\tilde{p}(\mathbf{z})$ is approximately close to the fixed distribution with easy geometry so that MCMC sampling is efficient.

Neural transport augmented samplers have been subsequently extended by Hoffman et al. (2019) who use more powerful flow architectures and, Jaini et al. (2021) who extend the idea to sampling from discrete probability densities using flows with surjective transformations (Nielsen et al., 2020). We believe these ideas and extensions for neural transport augmented samplers can also be used to sample from an invariant density $\pi$ by defining the flow transformations to be equivariant á la Köhler et al. (2020).

# B Related Work

In this paper, we proposed equivariant Stein variational gradient descent algorithm for sampling from densities that are invariant to symmetry transformations. Another contribution of our work is subsequently using this equivariant sampling method to efficiently train equivariant energy based models for probabilistic modeling and inference. Perhaps the closest work to that presented in this manuscript is that of Liu and Wang (2017) who first[3] used SVGD for training energy based models. However, their work does not consider incorporating symmetries in to either the sampler or the energy model itself.

Separately, a major contribution of our paper is indeed extending SVGD to incorporate symmetries present in the underlying target density. Since it was introduced by Liu and Wang (2016), Stein variational gradient descent has garnered a lot of attention as an alternative to Monte Carlo methods for sampling in Bayesian inference problems courtesy of its flexibility and accuracy obtained by combining variational inference and sampling paradigm. Stein variational gradient descent has been subsequently extended by Wang et al. (2019) to incorporate geometry information using matrix valued kernels, and to discrete spaces in Han et al. (2020). While, Duncan et al. (2019) have studied the convergence properties of Stein variational gradient descent under mean-field convergence analysis, a thorough theoretical understanding is still lacking in finite particle limit. Several works have, however, empirically probed limitations of Stein variational gradient descent. Particularly, Stein variational gradient descent is susceptible to collapsing to a few modes depending on the initial configuration of the particles (Zhang et al., 2020; D'Angelo and Fortuin, 2021). As we discussed towards the end of Section 3, incorporating symmetries alleviates this problem partially when the group factorized distribution is unimodal. Furthermore, the problem of mode collapse in Stein variational gradient descent can be addressed by either adding noise to the SVGD update (cf. Equation (3)) or using an annealing strategy as proposed in D'Angelo and Fortuin (2021).

Another contribution of this paper is learning equivariant Energy-Based Models using equivariant Stein variational gradient descent. Energy Based Models have witnessed a revival recently. The primary difficulty in training energy based models is the need to evaluate the partition function which is often intractable,. Thus, training energy based models require methods that can approximate this partition function. One line of ideas thus is to train an auxiliary sampling network that generates samples to approximate the partition function (Kumar et al., 2019; Xie et al., 2018) making it eerily similar in essence to Generative Adversarial Networks (GANs) (Finn et al., 2016). However, as discussed in Du and Mordatch (2019), such strategies are prone to mode collapse since the sampling network is often trained without an entropy term.

An alternative to this is to use Markov Chain Monte Carlo Method to directly estimate the partition function providing several benefits afforded by MCMC sampling methods. This idea was first proposed by Hinton et al. (2006), termed as Contrastive Divergence algorithm, which used gradient

---

[3]As far as the authors are aware.

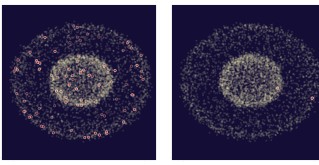
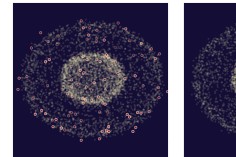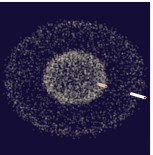

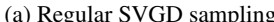

| (a) Regular SVGD sampling | (b) Invariant SVGD sampling |

Figure 8: *Recommended to view in color*. For both regular (8a) and invariant (8b) SVGD from left to right the original samples, and the samples projected on the group-factorized space are shown. Translucent yellow dots represent the distribution.

free MCMC chains initialized from training data to estimate the partition function. This was subsequently extended by Tieleman (2008) who introduced *persistence* in contrastive divergence where a single MCMC chain with a persistent state is employed to sample from the energy model. However, there are problems still with training EBMs using contrastive divergence. Specifically, EBMs training with contrastive divergence may not capture the target distribution faithfully since the MCMC chains used while training are truncated that lead to biased gradient updates hurting the learning dynamics Nijkamp et al. (2019); Schulz et al. (2010). Towards this end, a sampling procedure that converges quickly to sample from the energy function will potentially help to train energy based models. Thus, modelling invariant densities using equivariant energy functions that are trained using samplers that incorporate the symmetries in the energy function will potentially help with the training of such models.

Equivariant Stein variational gradient descent provides an efficient sampling procedure to train equivariant energy based models. However, modelling the equivariant energy function itself is mostly due to the tremendous advances in geometric deep learning (Bronstein et al., 2021). Particularly, we can leverage the various proposed architectures that that incorporate symmetries to model an equivariant energy function. In our experiments, we utilized these advances to model rotations using steerable CNNs (Weiler and Cesa, 2019), and molecules using E(n)-equivariant graph neural nets (Satorras et al., 2021).

## C Equivariant Stein Variational Gradient Descent

In this section, we provide the details for the toy experiments presented in Section 3 as well as additional experiments and plots.

### C.1 Additional SO(3) concentric spheres

In Figure 8, we extend the experiments presented in Section 3 to a distribution invariant to the $SO(3)$ symmetry group. The results further support the observations discussed in Section 3 that for continuous symmetry groups equivariant SVGD is able to capture the target density more faithfully due to its ability to model long range interactions whereas vanilla SVGD collapses particles to a single orbit representing the high density region in the probability landscape.

### C.2 Experimental setup

$C_4$-**Gaussians** The target $C_4$-Gaussians distribution for which we use equivariant SVGD to draw samples from is defined as a mixture of four Gaussians with uniform mixing coefficients. The mean of each Gaussian is located at a radius of 3 and they all have a covariance of $[1, \frac{1}{5}]$. The 50 starting samples are drawn from a two-dimensional Normal distribution with the mean at the origin and a covariance of $[2, 2]$. The samples are transformed over 25000 SVGD steps with a step size of 0.02. SVGD uses a scalar RBF kernel with a bandwidth of 0.2 for both regular and equivariant sampling.

**Concentric circles** The inner and outer circle for the concentric circle example are located at a distance of 4 and 8 from the origin respectively. A normal distribution with variance 0.5 describes the width of the concentric circles. The starting distribution is given by the two-dimensional uniform

distribution in the range $[-8, 8)$ for both dimensions. The RBF kernel used for SVGD has a bandwidth of 0.005. All other SVGD settings are kept consistent with the $C_4$-Gaussians example.

**Concentric spheres** The concentric spheres toy example is setup in a manner very similar to concentric circle experiment. Specifically, the target distribution is parameterized by the radius of the two spheres and the variance of the Gaussian distribution for the width of the spheres. In this instance, the two spheres have a radius of 4 and 9 and the two Gaussian distributions have a variance of 0.3. Again, similarly the starting distribution is given by uniform distribution in the range $[-8, 8)$. However, this time it is a 3-dimensional distribution. The RBF bandwidth is set to 0.001 and a total of 100 samples are drawn.

# D Equivariant Joint Energy Models

Here, we present experimental details for the toy example presented in Section 4, additional plots for Figure 4, as well as an additional toy experiment for training equivariant joint energy model using equivariant Stein variational gradient descent.

## D.1 Equivariant JEM trained with vanilla SVGD

In Figure 9, we continue the experiment presented in section Section 4. In this figure we provide the results for three models, namely: Regular energy based model trained with vanilla SVGD, equivariant energy based model trained with vanilla SVGD and, equivariant energy based model trained with equivariant SVGD. We find that in comparison with the non-equivariant EBM, an equivariant EBM (irrespective of the sampler used) better approximates the target distribution allowing the model to capture all four modes of the outer distribution while the non-equivariant EBM is unable to spread to those regions. However, compared to the equivariant-EBM trained with equivariant SVGD, we find that an equivariant-EBM trained with vanilla SVGD requires more training steps to reconstruct areas of low probability. This is due to the equivariant SVGD exploring a wider area of the landscape in the negative samples for the contrastive divergence algorithm since vanilla SVGD generates multiple samples in the same orbit due to only being able to capture local interactions.

## D.2 Additional JEM concentric circles

In Figure 10, we present the results of training an EBM on samples drawn from the concentric circles toy distribution. This is done for the same three combinations of EBM and SVGD sampler as before: a non-equivariant EBM trained with regular SVGD, an equivariant EBM trained with regular SVGD and an equivariant EBM trained with equivariant SVGD. We find that the trained EBMs show the same results as observed in the JEM trained on the $C_4$-Gaussians. A non-equivariant EBM is by far the worst of the three combinations. It specifically has a hard time in reconstructing the outer-ring. Stepping up to a equivariant EBM does improve on this aspect as a slight uptick can be seen at the location of the outer-ring. However, to also fully capture the areas of low probability, the regular SVGD has to be replaced by our equivariant version.

## D.3 Experimental setup

The experimental setup for the experiments with regular EBMs trained using vanilla SVGD consists of three parts: 1) defining the target distribution and the construction of the training dataset, 2) defining the energy model and the training parameters and, 3) defining the SVGD kernel and the sampling parameters. Note that for both the $C_4$-Gaussians and the concentric rings experiment the setup is kept consistent between the three combinations of EBM and SVGD method. Constructing the equivariant representations for the equivariant EBM models does not add additional parameters to the models.

$C_4$ **Mixture of Gaussians** The target distribution is defined as two sets of $C_4$-Gaussians, one at a distance of 7 from the origin and the other at a distance of 15. The inner-set of four Gaussians represents the distribution of the first class (i.e. $\pi(\boldsymbol{x}|y = 0)$) while the outer-set represents the second class (i.e. $\pi(\boldsymbol{x}|y = 1)$). Using this definition, both the class-conditional probabilities as well as the

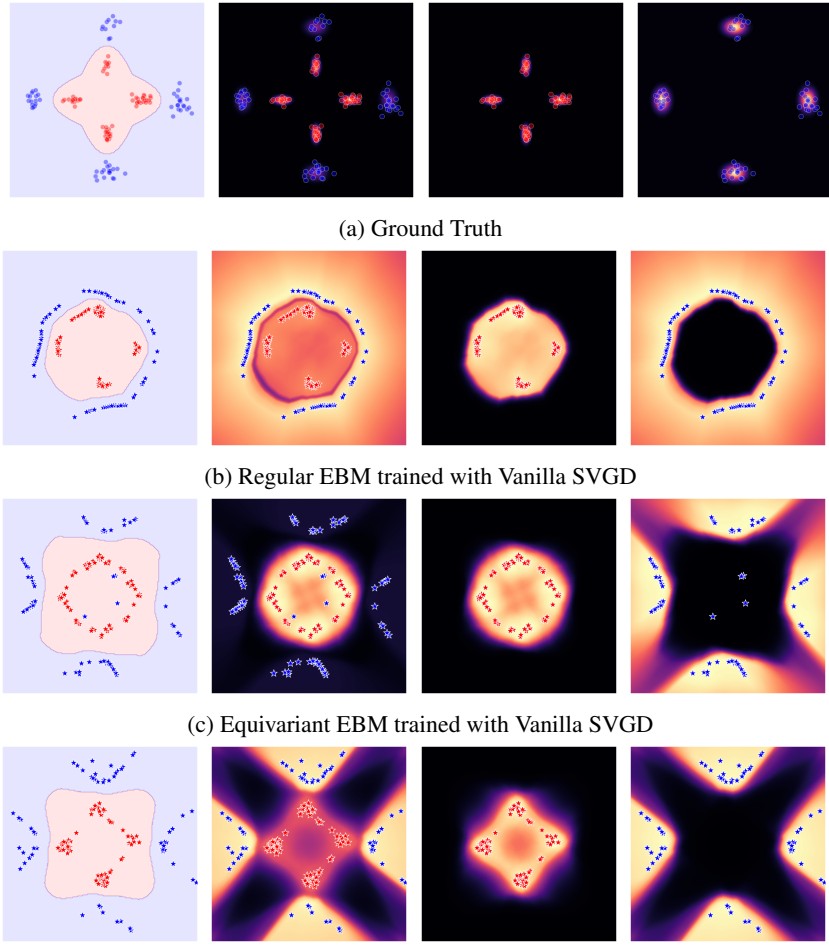

(a) Ground Truth

(b) Regular EBM trained with Vanilla SVGD

(c) Equivariant EBM trained with Vanilla SVGD

(d) Equivariant EBM trained with Equivariant SVGD

Figure 9: Extended version of Figure 4 with equivariant model and regular SVGD. *From left to right*: decision Boundary, samples and energy of joint distribution $\pi(\boldsymbol{x}, y)$, samples and energy of conditional distribution $\pi(\boldsymbol{x}|y = 0)$, samples and energy of conditional distribution $\pi(\boldsymbol{x}|y = 1)$. *Row 1:* Target distribution. *Row 2:* Non-equivariant EBM trained with vanilla SVGD. *Row 3:* E-EBM trained with E-SVGD. *Row 4:* E-EBM trained with vanilla SVGD.

joint distribution are invariant to the $C_4$ symmetry group. The dataset used for training the energy model contains of 128 samples equally divided amongst the two classes.

The regular EBM is defined as a 6 layer MLP with ReLU activation functions. The layers have 32, 64, 64, 64, 32, and 2 output nodes respectively with an input dimension as 2. The energy model is trained over 500 epochs with a fixed learning rate of 0.001 using the Adam (Kingma and Ba, 2014) optimizer with a batch size of 32.

The SVGD kernel used for sampling the 32 negative samples for the contrastive divergence step uses an RBF kernel with a bandwidth of 0.1. Each sampling step does 10,000 steps of SVGD with a step size of 0.9. We consider the SVGD to have converged if the norm of the update of the samples between two consecutive SVGD steps is less then $10^{-4}$. Additionally, the sampling uses persistence (Tieleman, 2008) with a 0.05 probability of resetting. When reset, new SVGD starting samples are drawn from the positive samples in the dataset. Furthermore, the positive samples in the dataset are used as additional repulsive forces by concatenating them to the batch of negative samples for calculating the update step in Equation (3).

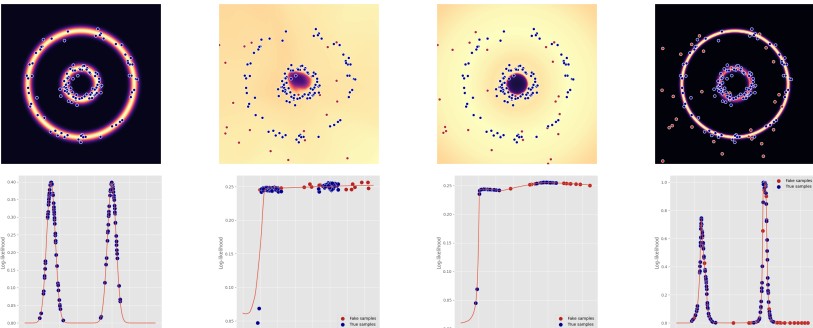

Figure 10: Visualization of (learned) distribution by EBMs trained concentric circles. Red dots represent the samples sampled during the last contrastive divergence step. Blue dots are the true samples used for training. *Row 1:* Two-dimensional visualization. *Row 2*: Samples projected on group-factorized spaces. *From left to right*: Target distribution, non-Equivariant EBM trained with vanilla SVGD, equivariant EBM trained with vanilla SVGD, equivariant EBM trained with equivariant SVGD.

**Concentric circles**   The concentric circles of the target distribution are located at a distance of 3 and 10 from the origin with equal mixing coefficients. This target distribution is used to sample a dataset with 128 training samples.

Both the energy function and most of the training setup are kept similar as for the $C_4$-Gaussians experiment. However, for this experiment a learning rate scheduler is used that reduces the learning rate at 150 and 400 epochs to 0.0005 and 0.0001 respectively. Additionally the Mean Square Error (MSE) loss is used as an additional supervision signal during training to demonstrate the use of loss given in Equation (8). The MSE loss is weighted by a factor of 0.5.

Negative samples are drawn using 10,000 SVGD steps with a bandwidth of 0.05. Additionally, we use a scheduler for the step-size for the Stein variational gradient descent algorithm which reduces the step-size at epoch 250 and 400 to 0.5 and 0.1 respectively. The use of persistence, its reset, and additional supervised SVGD repulsive forces is consistent with the $C_4$-Gaussians experiment.

# E   Experiments

In this section, we present the details for our experiments presented in Section 5.

## E.1   Many-body Particle System (DW-4)

**Experimental Setup:** The dataset for the Double-Well with 4 particles (DW-4) experiment is constructed by sampling a single example from each of the five meta-stable state configurations for the DW-4 potential. Each of these samples is then duplicated 200 times for a total of a 1000 training samples. A small amount of Gaussian noise is added to each sample to make them unique.

The EBM used to reconstruct the potential is parameterized by a 3 layer MLP with 64, 64, and 1 output nodes respectively. The input of the MLP is 8-dimensional (one for each coordinate of the 4 particles). Except for the final layer the ReLU activation function is used after each layer. For the final layer we use the activation function $\log(1 + x^2)$, where $x$ is the output of the final layer. The EBM is trained over 50 epochs using the Adam optimizer and a fixed learning rate of 0.01. The batch-size is 64.

We use scalar RBF kernels for all SVGD variant but depending on the type of SVGD used we use a different kernel bandwidth for the DW-4 experiment. Specifically, for regular SVGD we use a bandwidth of 0.1 and for equivariant SVGD we use 0.001. The influence of the RBF kernel bandwidth on the final results is discussed more in the next section. For each batch of negative samples, we evolved SVGD for 5,000 time-steps with a step size of 0.1 using the dataset samples as repulsive force. Persistence was used with a reset probability of 0.10. When reset, the starting coordinates for the DW-4 particles are independently sampled from the uniform distribution in the range $[-5, 5)$.

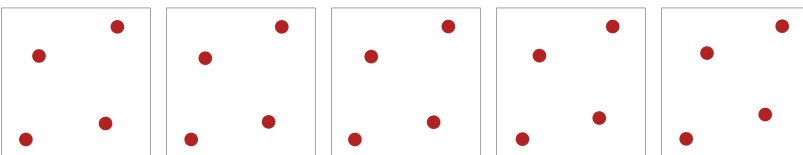

Figure 11: Training samples from the original dataset provided in Köhler et al. (2020). All samples represent the same meta-stable state and only differ by the addition of some Gaussian noise to the particle locations.

**Differences with Köhler et al. (2020):** As mentioned in the main paper, the experiment performed using the DW-4 is a slight deviation from an earlier proposed experiment using the same DW-4 potential in Köhler et al. (2020). In this earlier work, the authors propose to investigate the capacity of their proposed density estimation method (equivariant flows) to recover unseen meta-stable states of the potential when given only access to a single meta-stable states. To clarify, we refer to all possible local minima of the potential that are equivalent under rotation, translation and permutation symmetry as being a single distinct meta-stable states. Using this definition there are a total of 5 distinct meta-stable states (see Figure 7) for the DW-4 potential.

While the results presented in the original paper shows great success, it is however our understanding that the presented results can not be due to the proposed equivariance constraint on the normalizing flow. Precisely, the equivariant flow density proposed by the authors is invariant with respect to permutation of particles, rotation of the system of particles around its center of mass, and translation of the entire system. Thus, given only access to one of the 5 distinct meta-stable states, the equivariant flow only learns to assign a high probability to particle configurations that belong to this same meta-stable states. The 4 other distinct meta-stable states can not be recovered from the single presented meta-stable state using the symmetry transformations that the flow is invariant to. As the density estimated by the EBM trained using our equivariant SVGD proposed in this work is also only invariant to these same symmetries it would face the same restrictions. Instead, we believe that the method recovers the other 4 meta-stable state primarily due to the additional spread we observed in the equivariant sampling process (see Section 3) and the addition of the Gaussian noise to the training example. By adding this small amount of noise, the density estimator can not collapse its density into the single training example. As a result, every possible configuration of the four particles ultimately has a small non-zero probability. Given enough samples, there is herefore a non-trivial chance of sampling other meta-stable states as well. The equivariant constraint on the normalizing flow further amplifies this as the spread of non-zero probabilities not only occurs around the single training example, but also around all symmetry transformations of it.

The hypothesis of the previous paragraph is substantiated by the experimental results presented in Figure 12. We find that if we train an equivariant EBM using equivariant SVGD on the dataset supplied in Köhler et al. (2020) (see Figure 11) instead of the one we constructed ourselves, the sampling procedure can be forced to replicate the same results as presented by Köhler et al. (2020). If we use equivariant SVGD to sample from the trained EBM with the RBF kernel bandwidth set too high, the variance in the samples becomes too large. Thus, with a sufficiently large number of samples, searching through these samples reveals that all distinct meta-stable states have accidentally been recovered. When we significantly reduce the bandwidth, the same number of samples can be drawn but the variance will be low enough to only recover symmetry transformations of the original meta-stable state in the dataset. The later is the expected results given the set of symmetries the estimated densities are invariant too.

### E.2 Molecular Generation using QM9

For the molecular generation experiments we limit the large QM9 dataset to constitutional isomers of $C_5H_8O_1$. This molecule was selected due to the relatively high number of constitutional isomers (35) in the dataset in combination with its low atomic charge (46). This allows for sufficient variation within the dataset while keeping the molecules small enough to easily visualize and interpret.

The equivariant EBM is created using an Equivariant Graph Neural Network E-GNN (Satorras et al., 2021) with 4 Equivariant Graph Convolutional Layers. Each layer has 64 units. All other model configurations are kept consistent with those deployed in Satorras et al. (2021) for the same dataset.

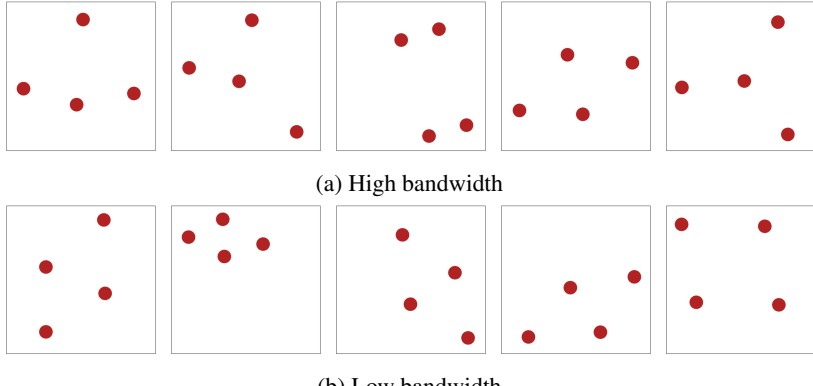

(a) High bandwidth

(b) Low bandwidth

Figure 12: Results of sampling using equivariant SVGD from an equivariant EBM trained with the dataset provided in Köhler et al. (2020). In fig. 12b a bandwidth of 0.0001 for the RBF-kernel is used while in fig. 12a a bandwidth of 2.0 is used. Note, sampling is done from the same EBM trained using equivariant SVGD with a bandwidth of 0.001.

The model was trained over 2500 epochs with a step wise learning rate scheduler. We start with a learning rate of 0.01 and reduce it to 0.005 and 0.001 at epochs 250 and 1,000 respectively. We used an RBF kernel with a bandwidth of 1 for equivariant SVGD which was evolved for 2,500 time-steps with a step size of 0.5. The persistent samples were reset with a 0.2 probability.

**Bond estimation**    Before visualizing the generated structures, we first post-process them to infer the bonds between the atoms. We do this by using the distance between atoms as a proxy for their probability of being bonded. In the following, we will describe this process in detail. We will rely on graph terminology where atoms are represented by nodes and bonds as undirected edges. The three steps for this process can be roughly described as: 1) bond all heavy atoms together such that they form a connected graph, 2) bond each hydrogen atom to one of the heavy atoms and, 3) create new bonds between atoms or double-up bonds between already connected atoms until each heavy atom has the required number of bonds.

As stated, the goal of the first step is to form a connected graph containing all the heavy atoms. We use the atom closest to the origin as the starting graph consisting of only this single node. From there on, we continuously add the atom that is closest to any of the atoms already in the connected graph. The atom newly added to the connected graph and the atom it was closest to are then connected by an edge/bond such that the graph maintains its connectivity. Note that when determining the distance to the connected graph for each atom, we only consider the distance to atoms that still have bonds available.

In the second step, we connect all hydrogen atoms to the connected graph of heavy atoms. We order this process based on the distance of each hydrogen atom to its closest heavy atom in the connected graph. In other words, we first calculate the distance from each hydrogen atom to all heavy atoms in the connected graph. Given all these distances we then iteratively bond the hydrogen atom that is furthest away from its closest heavy atom in the graph that still has bonds available to this heavy atom. After each newly connected atom pair, we update the number of bonds available for each heavy atom before continuing.

In the last step, we spend all remaining bonds available for the heavy atoms. To do this we repeat the following process until all bonds are spend. First, find the pair of heavy atoms that still have bonds left that are closest together. Second, find the closest pair of atoms that have bonds left and are not yet connected. If the distance between the second pair is not further than 1.1 times the distance between the first pair, then bond the second pair. Otherwise, connect the first pair.

In Figure 13 we show a full batch of sampled molecules. We find that in addition to the ones presented in the main body of the paper, most other molecules are also anecdotally correct. However, we do also find some weird structures. These can be roughly categorized in two classes. The first class contains faulty molecules that result from the sampling procedure. If we label the rows by the letter

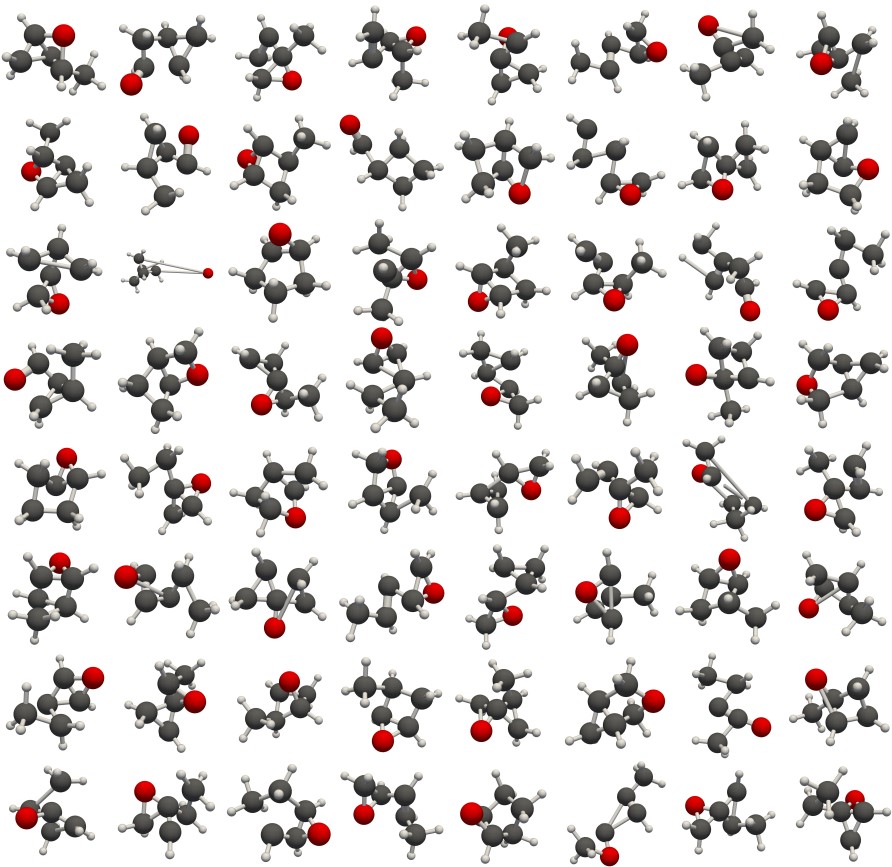

Figure 13: Full batch of 64 molecules sampled from the trained EBM. Note that this number is larger then the total number of molecules trained on.

A till H and the columns by the numbers 1 till 8, the molecule in C2 is one such example. The second class contains all molecules that show weird bonds. Molecule D7 and E3 are clear examples of this.

### E.3 Joint and Conditional Generation for FashionMNIST

We use the FashionMNIST dataset (Xiao et al., 2017) for this experiment with no data augmentation i.e. all the models are trained with images in their natural orientation. The test set is however preprocessed to contain images that have been randomly rotated using the $C_4$ symmetry group i.e. rotation angles from the set $\{0°, 90°, 180°, 270°\}$. The equivariant energy based model is created using a $C_4$-steerable CNNs (Weiler and Cesa, 2019) consisting for eight $C_4$-steerable convolutional layers followed by a group pooling layer and a fully connected layer. The regular EBM consists of the same architecture wherein the individual layers are not equivariant i.e. the steerable CNNs are replaced by normal CNN layers. We train all the models using the joint-energy model loss described in Equation (8). We train the models using both equivariant SVGD and vanilla SVGD resulting in three combinations of models, namely: Equivariant EBM trained with equivariant SVGD, equivariant EBM trained with vanilla SVGD and, regular EBM trained with vanilla SVGD.

We trained each model for 300 epochs using the Adam optimizer (Kingma and Ba, 2014) using a batch size of 64. For equivariant SVGD, we evolve the dynamics for 1,000 time-steps per mini-batch but due to slow convergence we had to increase this to 3,000 time-steps for vanilla SVGD. We used an RBF kernel for SVGD with a bandwidth of 0.1 for vanilla SVGD and 0.005 for equivariant SVGD with a step size of 0.08. We also used persistence with a reset probability of 0.1.