# OpenReview forum: "Learning Equivariant Energy Based Models with Equivariant Stein Variational Gradient Descent"
_NeurIPS.cc/2021/Conference — NeurIPS 2021 Poster_

### Official Review · Reviewer_FRDu · 2021-07-13

**Rating:** 6
**Confidence:** 4

**Summary:**

This paper worked on efficient sampling from distributions that have special structures regarding symmetries. Focusing on the group invariance of the target distribution, the authors developed a new SVGD algorithm, of which kernel function incorporates the group invariance. Various numerical experiments show the superior efficiency of the new SVGD compared to naive SVGD.

**Limitations And Societal Impact:**

Yes, they are shown in the checklist.

**Main Review:**

Overall, I think the paper provides practical and useful extensions for SVGD, but some practically and theoretically important points are still missing to publish this work.

# Pros
-  This paper provides a new SVGD using a $\mathcal{G}$-invariant kernel function and its superior performance is supported in various tasks.

# Cons
-  The discussions on previous works should be presented in the main paper, otherwise, it is hard to follow for a non-expert why this work is important.
-  My major concern is the theoretical and numerical properties of $mathcal{G}-$invariant kernel is unclear. I will comment on it below.
- Another concern is the hyperparameter tuning in the kernel function is not clear.  In vanilla SVGD, we can adaptively tune the bandwidth of the RBF kernel using the median trick (although the authors used the constant bandwidth in numerical experiments).

# Comments and Questions:
- I am not sure that using the $\mathcal{G}-$invariant kernel provides a valid gradient flow and it can converge successfully. I think the geometry of the proposed dynamics is significantly changed from the vanilla SVGD due to the $\mathcal{G}-$invariant kernel,  so authors should discuss it, for example, considering the following points.
In previous works for SVGDs, to ensure the convergence, we often assume the translation-invariant kernel function, $k(x,y)=k(x-y)$. (for example, On the geometry of Stein variational gradient descent (A. Duncan, et al)). When using RBF kernel, this is satisfied but when we extend RBF kernel to $\mathcal{G}-$invariant kernel, it is not satisfied. Another common assumption is the eigenvalues of the kernel function (SVGD as a kernelized Wasserstein gradient flow of the chi-squared divergence, (S Chewi, et al)). Assumptions used in (A Non-Asymptotic Analysis for Stein Variational Gradient Descent (A Korba et al)) is that the norm of the derivative of kernel function is upper bounded,  ($|\nabla k(x,y)|\leq B$)  and the larger the constant $B$ is, the slower convergence we obtain. Interestingly, this is contradicted with the numerical experiment in Figure 1 on page 5 in the paper. I think the norm of the derivative of the  $\mathcal{G}-$invariant kernel function based on the RBF kernel is larger than the original RBF kernel.



- Authors should describe what kernel function is used in numerical experiments in the main paper although it was written in Appendix currently. Moreover, as the authors mentioned in Section 6, tuning-hyperparameters in kernel function is very important for SVGD. Thus, the authors should mention how the bandwidths are tuned in numerical experiments.
- Related to the bandwidth tuning, the authors described that bandwidths were fixed constant during the optimization in Appendix E. I think previous works including the original SVGD paper uses the median trick to tune bandwidths. As far as I know, using the median trick to adaptively tune the bandwidth during the optimization seems very important numerically. I strongly recommend running experiments not using constant bandwidth but uses the median trick for the bandwidth tuning for SVGD. However, I am not sure how to use the median trick for $\mathcal{G}-$invariant kernel, especially for continuous groups since it has the integral term and there are infinitely many "virtual" particles.





**Time Spent Reviewing:**

3 hours

---

> ### Author Response · Authors · 2021-08-09
> **Response to Reviewer FRDu**
>
> We thank the reviewer for their time and valuable comments. We address each of the concerns raised by the reviewer below.
>
> **Main contributions**: We’d like to emphasize that the major aim of the paper was to incorporate symmetries through equivariant transformations in probabilistic modelling. We achieved this goal through the following two major contributions of the paper:
> - **Equivariant SVGD**: By incorporating symmetries through equivariant kernels that results in an equivariant vector field we extended SVGD to Equivariant SVGD that ensures that evolves a set of particles through a dynamical system in which the density at each intermediate time step is $\mathcal{G}$-invariant.
>
> - **Equivariant Energy Models**:  However, the utility of such an equivariant sampler is restricted to problems where we have access to the invariant density $\pi$ (e.g. Bayesian posterior sampling). Thus, a second major contribution of the paper is to extend the utility of E-SVGD to train equivariant EBMs for problems where we are only given access to samples. We achieved this by extending EBMs and Joint Energy Models to incorporate symmetries through equivariant neural networks as energy functions for learning invariant densities when given access to only data. We make the training of equivariant EBMs efficient by utilizing the equivariant SVGD sampling and support this with experiments both on toy datasets and diverse real-world datasets.
>
>
> **Theoretical foundations for Equivariant SVGD**: The reviewer expressed concern about the theoretical and numerical properties of a $\mathcal{G}$-invariant kernel. We agree with the reviewer that studying the theoretical properties of Equivariant SVGD (convergence analysis, gradient flow formulation etc ) is important.
>
> However, we also believe such an analysis will be highly non-trivial and is best explored as a separate work (and is also made difficult here by the page-limit restrictions). Research in SVGD has been fueled by demonstrating its empirical performance in various tasks like Bayesian inference, learning deep probabilistic models, reinforcement learning etc which has motivated several additional works studying its theoretical properties.
>
> Our work here firmly lies in the first category of empirical demonstration of SVGD where we extend SVGD to incorporate symmetries and through the use of various toy examples and real-world problems demonstrate its effective performance both in modelling and sampling from invariant densities but also as a tool to efficiently train equivariant energy models.
>
> We thus believe that such a theoretical analysis is best explored in a future work with a dedicated manuscript to such an exposition. However, we address the main concerns raised by the reviewer which are two-fold: whether the use of an invariant kernel results in a valid gradient flow and secondly, the finite-time convergence of Equivariant SVGD.
>
> - **Gradient Flow**: We do not think that using a $\mathcal{G}$-invariant kernel does not lead to a valid gradient flow. Informally, we believe this following the work of [Liu, 2017] where the original SVGD algorithm is viewed as gradient flow of the KL divergence functional (section 3.4). Our formulation with an equivariant vector field $\Psi$ would follow the same argument as present there to view the resulting Equivariant SVGD as a gradient flow.
>
> - Secondly, we also note our equivariant SVGD also satisfies the analysis presented by [Lu et.al, 2018] in the large particle limit (which only requires the kernel to be symmetric and positive definite) and should enjoy similar convergence guarantees as SVGD in such a regime.
>
> - **Assumptions not satisfied**: The reviewer also mentioned that the use of an equivariant kernel does not satisfy some of the assumptions made in previous works [Duncan et.al, Korba et.al, Chewi et al.]. However, we do not believe that the assumptions made in these previous works are necessary for an SVGD algorithm to enjoy desired properties like convergence etc. The assumptions made in these works are geared to facilitate the theoretical exposition presented in each of these works but not satisfying these assumptions does not directly imply that the resultant procedure lacks any of these properties. Specifically,
>     - While an equivariant kernel is not translation invariant, [Duncan et.al] mention themselves that this assumption is not necessary and can be weakened easily but they chose to impose it for ease of presentation (Remark 4 before Section 4).
>
>     - Similarly, from our understanding from [Chewi, et.al], the restrictions on the eigenvalues on the kernel are only realised in the idealized setup when the kernel $K_{\pi}$ is the identity operator (section 4 beginning). Furthermore, this restriction is required only for an approach that proves finite-time convergence in the spirit of Theorem 1 in the paper. This led them to propose Laplacian Adjusted Wasserstein Gradient Descent which follows the analysis of Theorem 1 and gives finite-time convergence guarantees for an alternate formulation of SVGD.
>
>     - We do not believe that Fig.1 is a contradiction to the bound presented in [Korba et.al]. The figure (and other experiments like the one on FashionMNIST) amply shows that if the sampler incorporates prior information about the target density in the form of symmetries, then the sampling procedure is particle efficient. We believe this is an expected phenomena which has also been observed in several papers on equivariant neural nets. We also believe that the bound presented in [Korba et.al] may not directly be applicable to our regime since it is agnostic to the geometry of the target density and how it is incorporated in equivariant SVGD.
>
> Nevertheless, we thank the reviewer for bringing up this point about the theoretical properties of the equivariant extension of SVGD. We will add a discussion section about this based on our remarks above in the revised paper.
>
> We also agree with the spirit of the reviewers feedback that a systematic study for such equivariant samplers is required to be able to theoretically back up the empirical evidence presented in the paper, which we believe is ideally explored in a separate manuscript.
>
>
> **Hyperparameters of the Kernel function**: For our experiments we used an equivariant kernel which was constructed using an RBF kernel. We selected the bandwidth of the kernel through a hyperparameter grid search and using our experience based on toy datasets and small initial runs on real-world datasets to tune the bandwidth.
>
> We tried Equivariant SVGD with an adaptive kernel bandwidth for our initial toy experiments but we found it to also be highly sensitive to the other hyperparameters like step-size etc which led to less than ideal convergence behaviour. Thus, we restricted ourselves to a fixed kernel bandwidth for our experiments which have been included in the paper. These experiments demonstrate the effectiveness of E-SVGD both as a stand-alone sampler and in training equivariant EBMs with a fixed kernel bandwidth.
>
> Additionally, the code that we submitted with the paper consists of options to use an adaptive kernel instead of a fixed bandwidth kernel as well. We also note that for the continuous kernel, we can instead use Monte Carlo approximation (c.f. Section 3, Example 2) that enables a direct use of the median trick for the adaptive kernel bandwidth.
>
> We will add this discussion about the choice of kernel and the hyperparameters chosen for the experiments in the main body.
>
> **Previous Work**: Due to space constraints, we had to unfortunately relegate discussion of previous work to the appendix. However, we will try to include some of this discussion in the background section of SVGD (Section 2) to provide more context on the contributions of the paper.
>
> We hope we have addressed the major comments of the reviewer and we’d be happy to address any other questions during the rolling discussion period.
>
>
>
> - [Liu, 2017] Stein Variational Gradient Descent as Gradient Flow
>
> - [Lu et.al, 2018] Scaling Limit of Stein Variational Gradient Descent: The Mean Field Regime
>
> - [Duncan et.al] On the Geometry of Stein Variational Gradient Descent
>
> - [Chewi, et.al] SVGD as a kernelized Wasserstein gradient flow of the chi-squared divergence
>
> - [Korba et.al] A non-Asymptotic Analysis for Stein Variational Gradient Descent

---

> > ### Author Response · Authors · 2021-08-23
> > **Thank you again for your comments and feedback and happy to address anymore questions**
> >
> > We thank the reviewer again for all their time in reviewing our paper and providing valuable feedback.
> >
> > We hope our replies addressed the reviewers concern but if any concerns remain or if there are any additional questions, please do let us know and we'd be happy to address them in a follow-up.
> >
> > Thanks!

---

> > > ### Comment · Reviewer_FRDu · 2021-08-26
> > > **Satisfied with the explanation**
> > >
> > > Thank you for the detailed explanation.
> > >
> > > I am satisfied with your explanation and I will increase the score.
> > >
> > > Best regards

---

### Official Review · Reviewer_FYs8 · 2021-07-16

**Rating:** 7
**Confidence:** 2

**Summary:**

The paper proposes an equivariant version of the Stein Variational Gradient Descent (SVGD) Algorithm for sampling from densities with symmetries. Using this sampling algorithm, the authors develop equivariant versions of energy-based models for learning invariant densities.

**Limitations And Societal Impact:**

The authors addressed the limitations and societal impact in Appendix F.

**Main Review:**

**Quality**: I checked the mathematical formulations and proofs in the paper, and to the best of my knowledge they seem to be correct.

**Clarity**: The paper is in general clear and very well-written. Section 2 on Preliminaries and Setup can benefit from having a few more bolded subsections that distinguish between different methods (e.g. original SVGD vs. generalized form). Section 4 on the Equivariant Joint Energy Model can also be made more clear in the same way by adding subsections (e.g. original energy-based model vs. joint energy model).

**Originality**: The equivariant extension of SVGD methods is novel, while it also seems to be relatively straightforward. Specific examples of applying equivariant energy-based models have been considered in concurrent work [1], though this paper is original in the sense that (i) it works on general energy-based models than a specific one, and (ii) it applies an equivariant sampling method within the energy model.

**Significance**: The idea of introducing equivariance in energy-based models is a novel one, and the authors did a good job in ensuring equivariance either in the sampling step or for the network. At the same time, the theory part of the paper is more or less a straightforward extension (i.e. adding equivariance everywhere) from my understanding, so I will be more interested in the experimental results given in the paper.

On a high level, I think the experimental results are comprehensive in showing the benefits of equivariant SVGD vs the original one for learning symmetries. However, I come from a normalizing flow background and I am not very familiar with the experimental behavior of SVGD as a sampling method in general. I wonder if the authors can clarify some of the following observations about experiments:

(1) In Figure 3(d), is there a good explanation for why the samples from vanilla SVGD all lie nearly exactly on the same orbit? While I do believe that equivariant SVGD may have samples that are more widely distributed across the width of the circle, it is a bit surprising to see vanilla SVGD converging to everything on the same orbit with literally no width in distribution.

(2) In Figure 1, is there a good explanation for why all the vanilla SVGD experiments converge consistently to something different from the ground truth?

**References**:

[1] Wu, J., Shen, T., Lan, H., Bian, Y., & Huang, J. (2021). SE (3)-Equivariant Energy-based Models for End-to-End Protein Folding. bioRxiv.

**Time Spent Reviewing:**

10

---

> ### Author Response · Authors · 2021-08-09
> **Response to Reviewer FYs8**
>
> We thank the reviewer for their time, and feedback. We address the reviewer’s questions and comments below:
>
> **Related work**: We thank the reviewer for bringing the work of [Wu et.al 2021] to our attention that also uses equivarant energy function (albeit restricted to the SE(3) group) for modelling molecular structures. However, this work was released publicly two weeks after the NeurIPS submission deadline and is thus as the reviewer pointed out concurrent to our work. As the reviewer also correctly mentioned, our work is more general since we can use any equivariant energy model as well as the fact that we also propose an equivariant sampling mechanism that makes the training for equivariant EBMs more efficient. Overall, we believe the concurrent work provides further backing on the benefits of incorporating symmetries in probabilistic models.
>
> **Experimental observations**: To answer the reviewers question we would like to refer to Section 3 (page 5) that discusses the long-range interaction property of the E-SVGD method. By incorporating equivariance into SVGD particles are not only interacting locally but also globally along their orbit. Without these long-range interactions vanilla SVGD tends to concentrate all particles in areas of high likelihood (as illustrated by Fig 3d). As a direct result, vanila SVGD converges to a higher log-likelihood (as illustrated in Figs 1 and 2). This issue with vanilla SVGD can not directly be solved by increasing the bandwidth of the RBF-kernel as this would repulse particles in all directions and not just in the factorized space. In the case of Fig 3, SVGD with a larger RBF-kernel bandwidth would have resulted in the particles roughly following a multivariate normal distribution. Theoretically, further increasing the number of particles used in SVGD could see the log-likelihood converging to the true distribution, but this is highly impractical.
>
> **Clarity**: The comments by the reviewer regarding the clarity of the work are very much appreciated. We will incorporate them in the final version of the paper.
>
> We hope that we have sufficiently clarified the questions posed by the reviewer. If there are any more questions remaining we would be happy to answer them during the rolling discussion period.
>
> [Wu et. al 2021] SE (3)-Equivariant Energy-based Models for End-to-End Protein Folding. bioRxiv.

---

> > ### Comment · Reviewer_FYs8 · 2021-08-29
> > **Re: Response to Reviewer FYs8**
> >
> > Thanks to the authors for the clarification that cleared my concern over experiments.
> > I have raised my score.

---

### Official Review · Reviewer_7bET · 2021-07-16

**Rating:** 7
**Confidence:** 4

**Summary:**

This paper aims to develop EBMs which respect certain symmetries in the underlying data distribution. This is accomplished by first developing an equivariant sampling method, equivariant SVGD, and then applying that to EBMs.

Section 2 provides theoretical background on equivariance and learning equivariant distributions.

Section 3 provides a thorough investigation comparing vanilla SVGD to equivariant SVGD. Figure 3 is immensely helpful in explaining the visual intuition of the method. It explains that because equivariant SVGD uses an equivariant kernel, it is better able to capture long-range interactions of particles. Particles can interact from a long distance if they are along the same orbit that SVGD will be invariant to. It is then demonstrated that this leads to far more efficiency in terms of sample complexity and log-likelihood.

Section 4 introduces equivariant EBMs. First, this requires using recent advances in geometric deep learning to develop high-capacity equivariant neural-network based energy functions. Since the energy is equivariant, they propose using the equivariant SVGD sampler to improve mixing of sampling, which would allow for more efficient learning from CD. They additionally propose training joint EBMs following Grathwohl et al. (2019). This formulation allows them to incorporate labels, and train the joint model by factoring the loss into a CD term and a CE term. This loss also allows for learning on unlabelled data, for example in semi-supervised learning. They demonstrate their approach on a simple 2-dimensional toy example, and show they model can generate better conditional samples and a better energy.

In section 5 they demonstrate their approach on a number of datasets.

**Limitations And Societal Impact:**

The authors have adequately addressed the limitations and potential negative societal impact of their work.

**Main Review:**

Describe the strengths of the work:

This paper develops a thorough theoretical background on equivariant learning and different approaches for learning equivariant distributions. It clearly explains its relation to prior work and develops detailed theory for equivariant SVGD.

Explain the limitations of this work:

The paper does not sufficiently disentangle the effects of using an equivariant EBM and sampling from it using equivariant SVGD. While there are experiments in the appendix that demonstrate this on toy 2-dimensional data, there aren’t experiments which attempt to investigate this question on larger scale datasets. For example, it would be interesting to see if training with E-SVGD improves the performance of the energy function at out-of-distribution detection or calibration.
It would also be interesting to compare the run-times of the approaches with SVGD or E-SVGD. does SVGD require more steps then E-SVGD for the same accuracy/sample-quality? How much longer is an E-SVGD step compared to an SVGD step?

Clarity:

Overall, the paper is written clearly. There are a few minor typos/fixes I suggest:
Introduce acronym Reproducing Kernel Hilbert Space (RKHS)
line 40 just use the acronym SVGD
line 65: implies to -> implies
line 171: use of an equivariant kernel

Reproducibility:

An anonymized github link is provided for code, and thorough details of data processing, architectures, and hyperparameters is given in the appendix.


**Time Spent Reviewing:**

4

---

> ### Author Response · Authors · 2021-08-09
> **Response to Reviewer 7bET**
>
> We thank the reviewer for their time, and feedback. We address the reviewer’s concerns below:
>
> **Disentangling effects of E-SVGD and Equivariant EBM**: In addition to the toy example presented in Fig 4 (and more detailed in appendix c.f. Fig 9), we performed a similar experiment for a real world dataset with the FashionMNIST experiment. Here we used accuracy as a proxy to disentangle the contribution of E-SVGD and incorporating symmetries in the EBM on the final trained model performance. Specifically, we compare the setting where both the energy model and SVGD are equivariant (red curve) with the settings when the energy model was equivariant but trained using vanilla SVGD (green curve) and finally with a vanilla energy model incorporating no symmetries trained using vanilla SVGD (blue curve). The results there show that incorporating symmetries in both the energy model and sampler is better (both in performance and convergence) than just an equivariant energy model trained with vanilla SVGD which are both significantly better than a vanilla energy model trained with vanilla SVGD.
>
> However, we agree with the reviewer’s comment that it would be interesting to study the effect of incorporating such inductive biases on OOD detection as a future work.
>
> **Run times**: We thank the reviewer for this comment. A major bottleneck in deploying SVGD with current resources is the sequential nature of the process. This, however, is a bottleneck in terms of run-time for both SVGD and E-SVGD. While E-SVGD indeed performs more computations than SVGD per step in evaluating the equivariant vector field using an equivariant kernel, this can effectively be parallelized.  As a result, we found the run-time per step for SVGD and E-SVGD to be similar.
>
> In fact, as E-SVGD requires significantly less steps to converge (c.f. Fig 1 and Fig 6, rightmost column), we found the convergence times of E-SVGD compared to vanilla SVGD to be orders of magnitude faster. We will add a plot for comparison for SVGD and E-SVGD run-times in the revision to make this point more explicit.
>
> **Typos and Writing**: We thank the reviewer for pointing the typos out and the suggestions. We will incorporate these in the revised manuscript.
>
> We hope we have addressed the concerns of the reviewer and we’d be happy to address any other questions during the rolling discussion period.

---

> > ### Comment · Reviewer_7bET · 2021-08-25
> > **Reply**
> >
> > Thank you for answering my questions and concerns. In particular, the discussion on run-times is quite interesting. Further experiments on disentangling the effects of E-SVGD and equivariant EBMs on larger scale problems could be interesting, but I would classify this as a “nice to have” and not a requirement. I will leave my score unchanged for now.

---

### Official Review · Reviewer_WvCB · 2021-07-18

**Rating:** 7
**Confidence:** 4

**Summary:**

This paper studies the incorporation of symmetries (of densities) into probabilistic models for efficient sampling and learning. It first proposes a method dubbed equivariant Stein variational gradient descent (SVGD), which uses work on equivariant, matrix-valued kernels to construct a set of particles along an optimal gradient path in order to sample from and approximate a target distribution -- which due to the exploitation of symmetry is able to handle long-range interactions. Next, recent work in equivariant and geometric deep learning is leveraged to build equivariant energy based models -- where the energy function is modeled by an equivariant network. These equivariant EBMs are trained using contrastive divergence while also using equivariant SVGD for sampling. A set of experiments, on toy data, molecular structure generation, and a vision task demonstrate the efficacy and sample efficiency of the methods proposed, and thus the utility of incorporating symmetry.

**Limitations And Societal Impact:**

Limitations are discussed in context of the unstable behaviour of contrastive divergence.

**Main Review:**

This paper builds off some recent work (such as by Kohler) on incorporating symmetries into probabilistic models. More specifically it is concerned with the problem of efficient sampling and learning of equivariant probability densities for this purpose. Towards this end, the paper makes a number of contributions. First. an equivariant version of the Stein variational gradient descent procedure (SVGD) is proposed. SVGD leverages some ideas from matrix-valued and equivariant kernels to construct a set of particles along an optimal gradient path. SVGD is then extended to invariant densities. Next, using recent work on equivariant deep learning, equivariant energy based models are proposed. These are trained by using contrastive divergence, with samples generated using SVGD. On a diverse set of experiments, the paper attempts to show the benefit of incorporating symmetry into such probabilistic models.

The paper is divided into four sections. The first gives a self-contained introduction, introducing the background, some of the recent work in the area and states the problem. The next introduces and formally specifies an equivariant sampler, followed by equivariant energy based models. The last section provides evidence for the benefit afforded by the approach (both in terms of the equivariant sampling, and equivariant EBMs).

The work of Kohler is used as a starting point. The problem is that given access to a set of i.i.d samples from a G-invariant density \pi, we want to approximate \pi. Prior work has approached this problem by means of an equivariant normalizing flow (which has the usual setup for normalizing flows but with G-invariance thrown in -- an initial, latent, G-invariant density is subject to a series of G-equivariant diffeomorphic transformations, so as to convert it into the target density). Efficient implementation of such a normalizing flow requires careful construction of an underlying dynamical system. While Kohler's work is referred to as a starting point and inspiration, it is not sufficient for sampling (at least in the way stated in the paper). The paper then turns to the Stein Variational Gradient descent procedure for this purpose, and considers the incorporation of equivariance in it, much like in the work of Kohler. The usual formulation of SVGD results in a system of ODEs which describes the evolution of particles as mentioned earlier. Furthermore, prior knowledge can be incorporated in the formulation of SVGD easily by considering kernels that reflect some geometric properties of the problem. Equivariant SVGD is achieved by simply using group equivariant matrix valued kernels. One of the nice things about equivariant SVGD, as compared to plain SVGD is that due to the fact that any particle can exert a force on any point in the neighborhood of any point in its orbit, it can model some long-range interactions better. The advantage of equivariant SVGD is demonstrated using some simple and standard(-ish) toy tasks, such as C4 gaussians and concentric circles. It is also demonstrated that invariant SVGD converges much more rapidly and with much fewer samples. The paper reports that vanilla SVGD does not converge even with 32 times the number of samples and 50 times more iterations. Next the paper extends energy based models to incorporate equivariance. They are trained with the usual CD procedure, with samples generated using equivariant SVGD.

The experiments comprise of a toy dataset where the task is to reconstruct the potential function of a many-body system trained using a few meta-stable states. Another experiment focuses on generating molecular structures (isomers of a particular molecule). Recently proposed E(n) equivariant GNNs are used as the underlying equivariant network, while equivariant SVGD is used for generating samples. Such experiments are certainly to be taken with a pinch of salt, however, whatever their limitations (and scale), we see that the equivariant EBM is able to recover correct 3D arrangement of bonds. No baseline comparisons are made, which is a weakness. Last is a familiar task on fashion-MNIST.

In summary: The paper presents a complete pipeline for incorporating symmetries into certain probabilistic models. For each of the steps, existing machinery (such as equivariant kernels, equivariance in EBMs) is cleverly used. Overall, the paper is well written, and makes a strong contribution. All the technical details seem correct and well fleshed out. While the experiments have significant room for improvement, I believe they are sufficient for corroborating the story of the paper.

Minor comments:

- Line 30: typo "the energy function is equivariant neural network" -> "the energy function is a equivariant neural network"?

- Line 65: typo "which implies to the following important result" -> "which implies the following important result"


**Time Spent Reviewing:**

3 hours

---

> ### Author Response · Authors · 2021-08-09
> **Response to reviewer WvCB**
>
> We thank the reviewer for their time, positive feedback and comments. We address the concerns raised by the reviewer below:
>
> **Baseline Comparisons and Experiments**: In the experiments we included in the submitted paper a major goal was to demonstrate that modelling symmetries both in the probability model through equivariant EBMs and in a sampler (to give E-SVGD) leads to efficient learning in terms of sample complexity and convergence times. Toward this end, we restricted ourselves to comparing Equivariant EBMs trained with Equivariant SVGD and alternates where we used vanilla SVGD and/or vanilla EBM on toy datasets and FashionMNIST to drive this point. Additionally, in our experiments for DW-4 we also found that a model without incorporating symmetries is unable to recover the other metastable states (c.f. Fig 7, column 2).
>
> Our main purpose with the experiment was to provide justification to the central claims made in the paper and as the reviewer also notes that the experiments corroborate the main story. However, if the reviewer has specific suggestions to improve the experimental section, we’d be happy to address them in the revision.
>
> **Typos**: Thanks for pointing these out. We will fix them in the revision.
>
> We hope we have addressed the concerns of the reviewer and we’d be happy to address any other questions during the rolling discussion period.

---

### Decision · Program_Chairs · 2021-09-27

**Decision:**

Accept (Poster)

**Comment:**

The paper proposes an equivariant version of the Stein variational gradient descent algorithm for sampling from densities with symmetries. Using this sampling algorithm, the authors developed equivariant versions of energy-based models for learning invariant densities. The reviewers all agree that this is an interesting paper with nice contributions. I recommend acceptance as poster.